# CONDITIONAL INVARIANCES FOR CONFORMER INVARIANT PROTEIN REPRESENTATIONS

## ABSTRACT

Representation learning for proteins is an emerging area in geometric deep learning. Recent works have factored in both the relational (atomic bonds) and the geometric aspects (atomic positions) of the task, notably bringing together graph neural networks (GNNs) with neural networks for point clouds. The equivariances and invariances to geometric transformations (group actions such as rotations and translations) so far treat large molecules as rigid structures. However, in many important settings, proteins can co-exist as an ensemble of multiple stable conformations. The conformations of a protein, however, cannot be described as input-independent transformations of the protein: Two proteins may require different sets of transformations in order to describe their set of viable conformations. To address this limitation, we introduce the concept of conditional transformations (CT). CT can capture protein structure, while respecting the constraints on dihedral (torsion) angles and steric repulsions between atoms. We then introduce a Markov chain Monte Carlo framework to learn representations that are invariant to these conditional transformations. Our results show that endowing existing baseline models with these conditional transformations helps improve their performance without sacrificing computational efficiency.

## 1 INTRODUCTION

The literature on geometric deep learning has achieved much success with neural networks that explicitly model equivariances (or invariances) to group transformations (Cohen & Welling, 2016; Maron et al., 2018; Kondor & Trivedi, 2018; Finzi et al., 2020). Among applications to physical sciences, group equivariant graph neural networks and transformers have specifically found applications to small molecules, as well as large molecules (e.g. proteins) with tremendous success (Klicpera et al., 2020; Anderson et al., 2019; Fuchs et al., 2020; Hutchinson et al., 2021; Satorras et al., 2021; Batzner et al., 2021). Specifically, machine learning for proteins (and 3D macromolecular structures in general) is a rapidly growing application area in geometric deep learning, (Bronstein et al., 2021; Gerken et al., 2021). Traditionally, proteins have been modeled using standard 3D CNNs (Karimi et al., 2019; Pagès et al., 2019), graph neural networks (GNNs) (Kipf & Welling, 2016; Hamilton et al., 2017), and transformers (Vaswani et al., 2017). More recently, several works Jing et al. (2020; 2021); Jumper et al. (2021); Hermosilla et al. (2021) have enriched the above models with neural networks that are equivariant (invariant) to transformations from the Euclidean and rotation groups. While equivariance (invariance) to the transformations in these groups are necessary properties for the model, unfortunately, they are limited to only capture rigid transformations of the input object.

However, these models may not yet account for all invariances of the input pertinent to the downstream task. And the transformations they are invariant to do not depend on the input. For instance, invariance to the Euclidean group restricts the protein representation to act as if the protein were a rigid structure, regardless of the protein under consideration. However, treating proteins as rigid structures may not be optimal for many downstream tasks. And different proteins may have different types of conformations (protein 3D structures with flexible side chains) (Harder et al., 2010; Gainza et al., 2012; Miao & Cao, 2016) . The existing rigid body assumption in protein representations may hurt these methods in datasets & downstream tasks that require protein representations to be invariant to a specific set of protein conformations. For example, for most proteins, regardless of their side chain conformation (as long as viable) under consideration – their protein fold class/ other scalar properties remain the same, their mutation (in)stability remains unaltered, protein ligand binding affinity (apart from changes at the ligand binding site) remain the same, etc.

In light of this limitation of current methods, a question naturally arises: *Is it possible to learn conformer-invariant protein representations?*

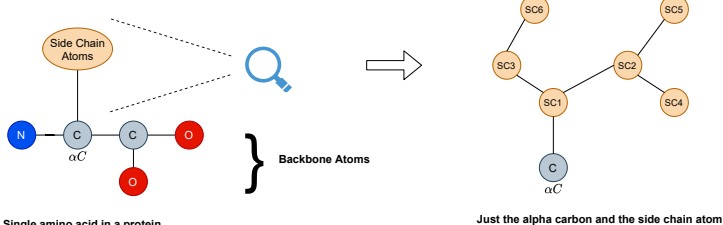

Figure 1: Magnified image of side chain of a single generic amino acid (here, with 6 atoms in the side chain) in a protein molecule. A protein molecule typically contains tens to hundreds of amino acids

**Our Approach:** We propose a representation method where the set of symmetries in the task are input-dependent, which we denote *conditional invariances*. For our specific application, we model every protein as a directed forest, where every amino acid forms a directed tree. We leverage the constructed directed forest of the protein to sample viable protein conformations (where viability is checked with Protein structure validation tools such as Molprobity (Davis et al., 2007; Chen et al., 2010)), which is additionally coupled with a Markov chain Monte Carlo (MCMC) framework to create data augmentations to train the neural network to learn conformer invariant representations.

Our **contributions** can be summarized as follows:
- We provide guiding principles for defining conditional invariant representations as inductive biases, which serves as a generalization of group invariant neural networks.

- We then provide a principled strategy to sample conformations for any given protein from the support of its protein conformer distribution (which consists of all its viable protein conformations) which captures their true flexibility. Viable conformations respect domain specific constraints such as those on dihedral angles, steric repulsions, among others. This is particularly useful in the scenario when the set of viable transformations is different across different proteins.

- Further, we develop an MCMC-based learning framework which guides the sampling of protein conformations such that the (asymptotic empirical) average representation obtained by all viable conformations of a protein are identical.

- Finally, we perform experimental evaluation of our proposal, where endowing baseline models with our proposed strategy (via an implicit data augmentation scheme) shows noticeable improvements on multiple different classification and regression tasks on proteins.

## 2 CONDITIONAL INVARIANCES FOR PROTEINS

The symmetry of an object is the set of transformations that leaves the object invariant. The notion of symmetries, expressed through the action of a group on functions defined on some domain, has served as a fundamental concept of geometric deep learning. In this section, we start by defining the concept of input-dependent *conditional symmetries* and then proceed to specifically tailor these conditional symmetries that are both computationally efficient and useful for representing protein conformations. Some group theoretic preliminaries are presented in the Appendix A.2.

**Definition 2.1** (Conditionally symmetric-invariant functions). A function $f : \Omega \to \mathbb{R}$, is said to be conditionally symmetric-invariant if

$$f(t_x \cdot x) = f(x) \quad , \forall t_x \in S_x, \ \ \forall x \in \Omega,$$

where $S_x$ is a set of transformations unique to element $x$ and $t_x : \Omega \to \Omega$.

It is easy to see that conditionally invariant functions are a generalization of group invariant functions where $S_x \equiv G \ \ \forall x \in \Omega$ where $G$ is a group. The above definition is motivated by the fact that representations for proteins may not necessarily be limited to be invariant just to group actions, but to a more general set of protein specific transformations. We detail this motivation next.

A protein molecule is composed of amino acids (say $n$ amino acids), where each atom in the protein belongs to one amino acid $\in \{1, \ldots, n\}$. Excluding the hydrogen atoms, every amino acid contains four atoms known as the backbone atoms (see Figure 1), and other atoms which belong to the side chain. Traditionally, protein structures are solved by X-ray crystallography or cryo-EM and the obtained structure is normally considered a unique 3D conformation of the molecule. Molecules available via the Protein Data Bank (Berman et al., 2000) generally include only unique sets of coordinates to demonstrate the 3D structures, which lead to the unique answer as 'gold standard' in structure prediction, such as protein structure prediction competitions - CASP (Kryshtafovych et al., 2019). Under these assumptions protein side-chain conformations are usually assumed to be clustered into rotamers, which are rigid conformations represented by discrete side-chain dihedral angles.

However, works over the past decade (Harder et al., 2010; Gainza et al., 2012; Miao & Cao, 2016) have observed that — while the backbone atoms of all $n$ amino acids in a protein molecule together (i.e. $4n$ atoms) form a rigid structure, its side chains can exhibit flexible (continuous and discrete) conformations beyond clustered rotamers. The underlying goal of our work is to capture this inherent flexibility of proteins and to ensure different conformers of the same protein get identical representations (More details – Appendix A.10). The above problem of capturing protein conformations is further compounded by the fact that protein conformations are often times unique to the protein - i.e. conformations exhibited by a protein are input (protein) specific. The desiderata, therefore is a model which outputs conformation invariant representations when the conformations (symmetries) exhibited by a protein varies across proteins when access to only a single protein conformer is available.

We denote a protein as a tuple $p = (V, \boldsymbol{X}_s, \boldsymbol{X}_p)$ (data from pdb files) where $V$ is the set of nodes (every atom is a node) in the protein, $\boldsymbol{X}_s \in \mathbb{R}^{m \times d}, d \geq 1$ is the scalar atom feature matrix associated with the atoms, $\boldsymbol{X}_p \in \mathbb{R}^{m \times 3}$ (where $m$ denotes the number of atoms in the protein) are the positional coordinates associated with the atoms in the protein. Without loss of generality, we number the nodes in $V = \{1, \ldots, m\}$ following the same ordering of the rows in $\boldsymbol{X}_s, \boldsymbol{X}_p$.

**Definition 2.2** (Rigid Backbone Protein Conformations). For an $m$-atom protein $p$ with atom positional coordinates $\boldsymbol{X}_p \in \mathbb{R}^{m \times 3}$ (recall that by definition $p$ contains information about $\boldsymbol{X}_p$), where $C_p \subset \mathbb{R}^{m \times 3}$ denotes the set of viable conformations of $p$, which keeps the positional coordinates associated with the backbone fixed. We use $T_p \subset \mathbb{R}^{m \times 3 \times 3}$ (where a 3x3 matrix is associated with every single atom) to denote the set of non-isometric transformations of $p$, which yield the set $C_p$ i.e. $\forall c_p \in C_p, \exists t_p \in T_p$, s.t. for an atom with index $i \in \{1, \ldots, m\}$, the new position of atom $i$ is $\boldsymbol{X}_p[i] t_p[i]^T = c_p[i]$, $\boldsymbol{X}_p[i] \in \mathbb{R}^{1 \times 3}$, $t_p[i] \in \mathbb{R}^{3 \times 3}$.

$T_p$ forms a set of transformations which acts on the protein atomic coordinates via matrix multiplication. Two elements of $T_p$ (with corresponding matrices of individual atoms and then aggregating for the protein as a whole) may or may not combine via matrix multiplication (as the binary operation) to result in another element of $T_p$ i.e. matrix multiplication is not necessarily closed in $T_p$. A concrete example of the non group structure of protein conformations is provided in Appendix A.3.

Unfortunately, sampling transformations from $T_p$ to obtain viable protein conformations is an unattainable task, since this would entail first sampling a transformation from $\mathbb{R}^{m \times 3 \times 3}$ and then verifying if the transformation is viable (and the vast majority of transformations are not viable). We will address this issue using an MCMC method - which is then combined with MCGD training of the neural network (Sun et al., 2018) to learn conformation invariant protein representations where different viable conformations are obtained via MCMC are used in every batch. Markov Chain Gradient Descent (MCGD) is a variant of Stochastic Gradient Descent (SGD), where the samples are drawn from the trajectory of a Markov Chain. We note that, while there have been prior Monte Carlo and MCMC based techniques for sampling protein conformations (Boomsma et al., 2013; Irbäck & Mohanty, 2006; Vitalis & Pappu, 2009), the overarching goals of the existing MCMC methods are significantly different compared to ours. The existing methods define a distribution over the conformations (based on the properties of the bonds, etc.) and then sample highly probable conformations from this distribution. Our method, is much simpler and only requires that the chain being used is ergodic and is invariant to the actual form of the distribution. Therefore, we devise a simple chain whose transitions are easier to sample and don't require us to compute conditionals of an energy based model. However, we also note that the existing Markov chains can be used as drop-in replacements to sample conformations as part of our framework. Before specifying our MCMC procedure, we specify the invariance requirements for learning protein representations.

A symmetric-invariant function for proteins (per Definition 2.1), should be invariant to (i) transformations which change its atomic coordinates to any of its rigid backbone conformations in $C_p$ (ii) transformations which change the atomic coordinates of all its atoms via rigid body transformations — group actions such as rotations/ translations (iii) a combination of the two above which transforms it into one of its viable conformations and is then further acted upon by rigid body transformations.

## 3 Obtaining Viable Conformations

Before we describe our MCMC procedure to sample atomic coordinates from the set of rigid backbone conformations , we need a way to sample from the local neighborhood of a conformation.

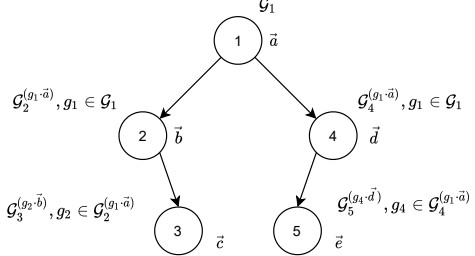

Figure 2: *Directed tree* corresponding to a set with 5 points which exhibits conditional invariances. Our proposed model (not limited to proteins), for example, allows node 2 and its descendants to transform its coordinates about node 1 (its parent) upon actions from group $G_2^{(g_1 \cdot \vec{a})}$, $g_1 \in G_1$. In practice, for protein molecules we use $G_1 = SO(3)$ and $G_i^{(\cdot, \cdot)} = SO(3) \forall i \neq 1$. We note that in protein molecules not all transformations about a node would be allowed due to steric repulsions between atoms as well as potential overlaps of atoms.

### 3.1 A SIMPLE PROTEIN CONFORMATION SAMPLING STRATEGY

Akin to Irbäck & Mohanty (2006), to efficiently sample a candidate conformation from $C_p$, we will follow a multi-step approach: First, we (i) construct a directed tree for every amino acid in the protein to capture the inherent flexibility in protein structures. Then, we (ii) leverage a directed forest of the protein (described next) to transform its atomic coordinates and check for viability.

**Part 1. Directed forest construction.** Using the input protein molecule, we construct a directed forest using the following three step procedure:

1. Each amino acid in the protein gets its own *directed tree*. The atoms in the amino acid's backbone form the base (root) nodes (i.e., there can be multiple base nodes in every amino acid). This is illustrated as node 1 in Figure 2's *tree*. The set of base nodes of all the amino acids in the protein are jointly referred to as the base nodes of the protein (or equivalently of the directed forest).

2. The atoms adjacent to the amino acid's backbone via a covalent bond become their immediate children in the *tree*. For example, the node SC1 forms the child of $\alpha$C in the Figure 1.

3. The other covalent bonds of the children establish the grandchildren of the root, and so on, until all the atoms in the amino acid are a part of the *tree*. For example, SC2 and SC3 become the children of the node SC1 in the directed tree constructed for Figure 1.

Figure 2 is an illustration of some directed tree constructed with the above procedure, where node 1 is the root node, with nodes $2, 4$ as its children and so on. It is important to note that, the above procedure ensures that in the constructed *tree*, each (non-root) node has a single parent and may have arbitrarily many children. Further, cycles/rings which are present in the molecule are broken on the basis of bond length, i.e., larger bonds are chosen before smaller bonds. Ties between same-length bonds are broken arbitrarily. While multiple directed forests can be created for a given protein since bonds of equal length are broken arbitrarily, in our implementation, given a protein, the tree is fixed because the directed forest for every protein is created exactly once (in the preprocessing step) - where the ties are broken deterministically based on their ordering in the "pdb file".

Next, we shall look at how the atomic coordinates are transformed using the directed tree.

**Part 2. Conditional transformations of atomic coordinates.** Let $\boldsymbol{X}_p$ be the available input atomic coordinates of the protein. To obtain a new candidate protein conformation (say $\boldsymbol{X}_p' \in \mathbb{R}^{m \times 3}$), we sample uniformly, one of the directed trees from the forest. Next, we transform the atomic coordinates of the subset of atoms in this directed tree. This set is denoted by $A \subset V$. That is, in the candidate conformation, $\boldsymbol{X}_p'[i] = \boldsymbol{X}_p[i]$, $\forall i \in V \setminus A$ — or equivalently $t_p[i] = \boldsymbol{I}$, $\forall i \in V \setminus A$.

Starting with the root node, we use breadth first search to traverse the directed tree and update all the descendants of the node currently being considered. For the backbone atoms, we leave $\boldsymbol{X}_p[i] = \boldsymbol{X}_p'[i]$ (or equivalently $t_p[i] = I$), i.e., atomic coordinates are unchanged for backbone atoms. Next, we use *pointed sets* to describe approximate transformations on a given protein through a directed tree .

**Definition 3.1** (Pointed sets). A pointed set is an ordered pair $(X, x_0)$, where $X$ is a set and $x_0 \in X$ is called the basepoint.

For instance, let $M_i = \{\boldsymbol{X}_p[j] : j \in N_i\}$, where $N_i$ is the set containing $i$ and all its descendants in its directed tree. Then, $(M_i, \boldsymbol{X}_p[i])$ is a pointed set.

Let the parent of node $i$ be denoted by $\circ$ and let $G_i^{(g_\circ \cdot \boldsymbol{X}_p[\circ])}$ (in practice, SO(3)) be the group from which actions $g_i \in G_i^{(g_\circ \cdot \boldsymbol{X}_p[\circ])}$ are sampled uniformly — that can transform the positional coordinates of node $i$ and its descendants about its parents in the directed tree, where $g_\circ \cdot \boldsymbol{X}_p[\circ]$ is the transformed atomic coordinates of the parent of node $i$ (we use BFS - so a top down approach). In the case of the root node (for example 1 in Figure 2), the actions are just drawn from $G_{root}$ (or $G_1$ in Figure 2).

The associated transformations of the atomic coordinates of the atoms in $N_i$ can be given by the mapping $h_i : (M_i, \boldsymbol{X}_p[i]) \to (\mathbb{R}^{|N_i| \times 3}, g_\circ \cdot \boldsymbol{X}_p[\circ] + g_i \cdot (\boldsymbol{X}_p[i] - \boldsymbol{X}_p[\circ]))$ where the coordinates of node $j \in N_i$ are updated as:

$$\boldsymbol{X}'_p[j] \leftarrow g_\circ \cdot \boldsymbol{X}_p[\circ] + g_i \cdot (\boldsymbol{X}_p[j] - \boldsymbol{X}_p[\circ]) \tag{1}$$

If the node $j \neq i$, its coordinates, i.e., $\boldsymbol{X}'_p[j]$ can be further updated as we traverse the tree. When $G_i^{(g_\circ \cdot \boldsymbol{X}_p[\circ])} = SO(3) \; \forall i \in V$, every node in the directed tree can be rotated about its parents as shown in Figure 3 (Appendix A.4) - which is the side chain of the amino acid shown in Figure 1.

---

**Algorithm 1** Sampling a viable conformation of $c_p$

---

1: **Input:** Conformation $c_p$ of protein $p$
2: Obtain the directed forest corresponding to the protein with $n$ directed trees where $n$ is the number of amino acids in the protein.
3: **repeat**
4:     Sample $y \sim \text{UNIF}(\{1, 2, \cdots, n\})$ and obtain the corresponding directed tree
5:     Traverse through directed tree via BFS and update atomic coordinates via Equation (1) to obtain $c'_p$ a candidate conformation of $c_p$— where group actions are always sampled uniformly from SO(3).
6:     Check if $c'_p$ is a valid protein conformation via protein structure validation tools
7: **until** $c'_p$ is a valid conformation
8: **Return:** $c'_p$, a valid conformation of $c_p$

---

However, each candidate $\boldsymbol{X}'_p$ obtained via the above transformation process may not belong to $C_p$. To that end protein structural validation tools may be used to check the validity of these candidates. Molprobity (Davis et al., 2007; Chen et al., 2010; Williams et al., 2018) is such a tool,which outputs scores corresponding to different metrics (such as number of dihedral (torsion) angles outside the allowed threshold, number of atom-atom clashes arising due to steric repulsions, etc) which can be compared with the originally provided conformation $\boldsymbol{X}_p$. Note, that while Molprobity is not perfect and we may end up developing models more invariant than just to viable conformations, alternatively (more precise) tools including those which allow for the evaluation of molecular force-fields, can be used to check validity. Additionally, the goal of this work is not to use Molprobity/ other protein structure validation tools as a part of generative models and a more invariant model does not necessarily affect performance on unseen but valid protein test data.

Our algorithm to sample viable conformations is summarized in Algorithm 1. The `repeat`-loop proposes a conformation in the *neighborhood* of the current conformation $c_p$ which is only accepted when the proposed conformation $c'_p$ is a valid conformation. As such it is an acceptance-rejection sampling algorithm. We will see later that the actual sampling probability distribution is not required to be known by our procedure. Our MCMC procedure defined next only requires Algorithm 1 to follow certain conditions which we argue it follows.

### 3.2 SAMPLING CONFORMERS VIA MCMC

We now describe the MCMC procedure that, starting at any conformer configuration $c_p^{(0)} \in C_p$ we will, in steady state, sample conformations according to a unique stationary distribution which depends on the protein $p$, where $C_p$ is the set of all viable conformations of $p$ as defined in Definition 2.2.

To that end we first define the transition kernel of the Markov chain. Algorithm 1 samples a conformation $c'_p$ in the *neighborhood* of a given conformation $c_p$ using a directed forest. The neighborhood $\mathcal{N}(c_p)$ of the conformation $c_p$ is loosely defined as the set of all possible conformations $c'_p$ which may result from running Algorithm 1 with $c_p$ as input. We denote the probability distribution induced by Algorithm 1 as the transition kernel $\kappa(c'_p|c_p)$ which has support $\mathcal{N}(c_p)$.

**Definition 3.2** (Conformer Sampling Markov Chain (CSMC) $\boldsymbol{\Phi}_p$). We define the conformer sampling Markov chain $\boldsymbol{\Phi}_p$ as a time-homogenous Markov chain over the state space $C_p$ with transition kernel

$\kappa$ as defined above, where $C_p$ is the set of valid conformations associated with given protein $p$ as defined in Definition 2.2.

The consistency of our learning procedure does not depend on the precise definition of $\kappa$. However, our procedure relies on the fact that any conformer $c_p' \in C_p$ can be sampled by $\kappa$ in a finite number of steps, starting from a conformer $c_p$. We use this fact to show that the chain $\mathbf{\Phi}_p$ converges to a unique stationary distribution regardless of the initial conformer. (All proofs in Appendix A.5).

**Proposition 3.3.** *Given the CSMC $\mathbf{\Phi}_p$ from Definition 3.2 whose transitions are governed by $\kappa$ which is implicitly defined by Algorithm 1, for any pair of conformers $c_p, c_p' \in C_p$, there exists $\tau_p < \infty$, independent of $c_p$, such that $P_{\mathbf{\Phi}_p}^{\tau_p}(c_p, c_p') > 0$, where $P_{\mathbf{\Phi}_p}^{\tau_p}$ is the $\tau_p$ step transition probability.*

Next, we show the existence of a unique steady state distribution of our Markov chain.

**Proposition 3.4.** *The CSMC $\mathbf{\Phi}_p$ defined in Definition 3.2 is uniformly ergodic if Proposition 3.3 is satisfied. Specifically there exists a unique steady state distribution $\pi_p$ such that for all $c_p \in C_p$, $\|P_{\mathbf{\Phi}_p}^n(c_p, \cdot) - \pi_p(\cdot)\| \leq \mathrm{C}\,\mathrm{R}^n$, where $\mathrm{C} < \infty$ and $\mathrm{R} < 1$ are constants that depend on $\mathbf{\Phi}_p$, $P_{\mathbf{\Phi}_p}^n$ is the $n$ step transition probability and $\| \cdot \|$ is the $\ell_1$ norm.*

Given that we now have the ability to draw samples from a Markov chain which achieves a unique stationary distribution $\pi_p$ on $C_p$, we shall leverage this next, in our learning framework to learn conformer invariant representations of proteins.

## 4 LEARNING FRAMEWORK

We shall employ a learning strategy, where we use the viable conformations obtained via the Markov chain to learn a function which outputs conformer invariant representations.

Let $\mathcal{D} = \{(x_j, y_j)\}_{j=1}^N$ be the input data (where we have a single conformer for every protein in the dataset). We shall consider a single data point $(x_j, y_j)$ henceforth, where $x_j = (V, \boldsymbol{X}_s, \boldsymbol{X}_p)$ is the input protein. We consider a supervised learning setting where $y_j$ is the associated target. We consider both classification and regression tasks.

Let $C_j = \{c_{j_i}\}$ be the set of viable conformations of protein $x_j$. We only consider viable conformations which defer only in the atomic coordinates matrix $\boldsymbol{X}_p$. For a given protein $x_j$, we shall use $x_{j_i}$ to denote protein $x_j$ but with $\boldsymbol{X}_p$ of the original protein modified by $c_{j_i}$, and use $S_j = \{x_{j_i}\}$ to denote the set of all viable $x_{j_i}$ i.e. the state space.

Let $f : \Omega \rightarrow \mathbb{R}^d$, $d > 0$ be any function with learnable parameters $\theta^f$, (for e.g. any neural network model such as 3D CNN, GNN, Transformer, LSTM, etc.) which takes a protein (in the form $(V, \boldsymbol{X}_s, \boldsymbol{X}_p)$) as input and outputs protein representations. The function $f$ need not necessarily output conformer invariant representations of the protein $x_j$. Then, a simple way to obtain conformer invariant representations of protein $x_j$ (apart from using trivial functions such as a constant function or function independent of $\boldsymbol{X}_p$) is computing its expected value over all its viable conformations. We shall denote $\overline{\overline{f}}$ to denote a function which outputs conformer invariant representations of a protein.

$$\overline{\overline{f}}(x_j; \theta^f) = \mathbb{E}_{X \sim \pi_j}[f(X; \theta^f)] \tag{2}$$

where $\pi_j(\cdot)$ is the steady state distribution of the Markov chain associated with the protein $x_j$. As such, $\overline{\overline{f}}(x_j; \theta^f) = \overline{\overline{f}}(x_j'; \theta^f)$ for any viable protein conformation $x_j' \in \mathcal{S}_j$. Subsequently, to learn an optimal $f$ (which we denote by $f^\star$), we wish to minimize the loss, defined as follows:

$$L(\mathcal{D}; \theta^f, \theta^\rho) = \frac{1}{N} \sum_{j=1}^N \hat{L}(y_i, \rho(\overline{\overline{f}}(x_j; \theta^f); \theta^\rho))$$

$$= \lim_{k \to \infty} \frac{1}{N} \sum_{j=1}^N \hat{L}(y_i, \rho(\frac{1}{k} \sum_{i=1}^k f(x_j^i; \theta^f); \theta^\rho)) \tag{3}$$

where $\hat{L}$ is a convex loss function e.g. cross entropy loss, $\rho$ is some differentiable function with parameters $\theta^\rho$ (in practice, an MLP) and $\lim_{k \to \infty} \frac{1}{k} \sum_{i=1}^k f(x_j^i; \theta^f)$ can be employed as an asymptotically unbiased and consistent estimate of $\overline{\overline{f}}$ where $x_j^i$ is the $i^{th}$ sample from the Markov chain corresponding to the protein $x_j$

Since Equation (3) is computationally intractable, we employ a surrogate for the loss given by:

$$J(\mathcal{D}; \theta^f, \theta^\rho) = \lim_{k \to \infty} \frac{1}{N} \sum_{j=1}^{N} \frac{1}{k} \sum_{i=1}^{k} \hat{L}(y_i, \rho(f(x_j^k; \theta^f); \theta^\rho)) \tag{4}$$

Observe that in Equation (4), the expectation over the conformations is now outside the $\hat{L}$ and $\rho$ functions while still remaining conformation invariant (however, the optimal parameters corresponding to minimizing $J$ are different from minimizing $L$). Following (Murphy et al., 2019b;a), we note that, when $\rho$ is the identity function, Equation (4) serves as an upper bound for Equation (3) via Jensen's inequality. However, learning representations by averaging over multiple conformations (in training) is still computationally expensive (back-prop and high variance issues). Next, we show how MCGD (Sun et al., 2018) can be leveraged to make this tractable (more details in Appendix A.10).

Making the case for simplicity, we next show convergence properties using the full batch setting. Note that is procedure and proofs are equally applicable in the mini-batch setting with minor modifications.

**Full-batch training:** We consider the data as a single batch $\mathcal{B} = \{(x_1, y_1), ..., (x_j, y_j), ..., (x_N, y_N)\}$ and for a data point $(x_j, y_j)$, denote the corresponding steady state distributions of the corresponding Markov chains using $\pi_j$ and the state space as $S_j$.

We denote all learnable parameters of $\rho \circ f$ as $\theta \equiv (\theta^f, \theta^\rho)$ and consider $\theta \in \Theta \subseteq \mathbb{R}^m$, $m > 0$ such that the function $\rho \circ f$ may be non-convex for some values of $\theta$.

We consider a sequence of step sizes $(\gamma_k)_{k=0}^{\infty}$ which satisfy:

$$\sum_k \gamma_k = +\infty, \quad \sum_k \ln k \cdot \gamma_k^2 < +\infty \tag{5}$$

and follow a gradient scheme where parameters are updated as:

$$\theta_k = \theta_{k-1} - \gamma_k \mathbf{Z}_k, \quad k > 0 \tag{6}$$

where $\mathbf{Z}_k = \frac{1}{N} \sum_{j=1}^{N} \nabla_\theta \hat{L}(y_j, \rho(f(x_j^{k-1}; \theta_{k-1}^f); \theta_{k-1}^\rho))$ and $x_j^{k-1}$ is the $k-1^{th}$ sample from the Markov chain corresponding to the data point $(x_j, y_j)$ and employ the Markov chain gradient descent procedure (Sun et al., 2018) where the loss in epoch $k, k > 0$ of training is obtained by

$$\hat{J}_k = \frac{1}{N} \sum_{j=1}^{N} \hat{L}(y_j, \rho(f(x_j^{k-1}; \theta_{k-1}^f); \theta_{k-1}^\rho)).$$

and minimizes the surrogate loss given in Equation (4) when k is sufficiently large so that the Markov chain has converged to its unique steady state distribution.

Next, we list the set of assumptions we make to ensure convergence of our conformer invariant MCGD procedure.

**Assumption 4.1.** We make the following assumptions:

1. For any $\theta \in \Theta$ and $x_j^k \in S_j$, the function $f$ is differentiable $\forall j$
2. $\sup_{\theta \in \Theta, x_j^k \in S_j} \{||\nabla_\theta \; \rho \circ f(x_j^k)||\} < +\infty$ i.e. the gradients are bounded.
3. $\forall x_j^k \in S_j, \forall \theta_1, \theta_2 \in \Theta, ||\nabla_{\theta_1} \; \rho \circ f(x_j^k) - \nabla_{\theta_2} \; \rho \circ f(x_j^k)|| < L||\theta_1 - \theta_2||$ for some $L \geq 0$ i.e., the gradients are $L-$Lipschitz.
4. $\mathbb{E}_{x_j^k \sim \pi_j}[\nabla_\theta \; \rho \circ f(x_j^k)] = \nabla_\theta \; \mathbb{E}_{x_j^k \sim \pi_j}[\rho \circ f(x_j^k)]$

Next, we formally state the convergence of our optimization procedure to optimal parameters $\theta^\star$ which yield conformer invariant protein representations.

**Proposition 4.2.** *Let the step sizes satisfy (5) and the function parameters $\theta$ be updated as (6) and Assumption 4.1 hold, then the MCGD optimization enjoys properties of almost sure convergence to the optimal $\theta$.*

The use of Markov chain gradient descent procedure to optimize the conformer invariant learning procedure optimizes the objective $J$ in Equation (4), and thus has the following implication on how outputs should be calculated at inference time:

Table 1: GNN(GCN), GVP-GNN, E(N) GNN - Baseline vs Conformation Invariant Strategies for multiple different tasks on proteins from the ATOM3D dataset. Corresponding to the metric, ↑ indicates that higher is better, while ↓ indicates that lower is better. Bold values indicate best results for a given row. The values for GNN were obtained from Townshend et al. (2020) and for the GVP-GNN from Jing et al. (2021). Gray colored cells indicates that the augmented model outperforms the baseline model.

| Task | Metric | Baseline (GNN[GCN]) | MCMC Augmented GNN (Ours) | Baseline (GVP GNN) | MCMC Augmented GVP-GNN (Ours) | Baseline (E(N) GNN) | MCMC Augmented E(N) GNN (Ours) |
|---|---|---|---|---|---|---|---|
| PSR | Global $R_s$ ↑ | $0.755 \pm 0.004$ | $0.761 \pm 0.004$ | $0.845 \pm 0.004$ | $\mathbf{0.852 \pm 0.006}$ | $0.827 \pm 0.004$ | $\mathbf{0.852 \pm 0.004}$ |
| LEP | AUROC ↑ | $\mathbf{0.740 \pm 0.010}$ | $0.672 \pm 0.012$ | $0.628 \pm 0.055$ | $0.704 \pm 0.039$ | $0.677 \pm 0.014$ | $0.714 \pm 0.005$ |
| LBA | RMSE ↓ | $1.570 \pm 0.025$ | $1.519 \pm 0.022$ | $1.594 \pm 0.073$ | $1.435 \pm 0.007$ | $1.392 \pm 0.001$ | $\mathbf{1.384 \pm 0.011}$ |
| MSP | AUROC ↑ | $0.621 \pm 0.009$ | $0.662 \pm 0.008$ | $0.680 \pm 0.015$ | $\mathbf{0.857 \pm 0.049}$ | $0.652 \pm 0.006$ | $0.843 \pm 0.037$ |

**Inference time:** We estimate $\bar{\bar{f}}$ using an empirical average of $f$ evaluated over conformations (in practice, average final layer [before softmax] representations) visited by the Markov chain :

$$\bar{\bar{\hat{f}}}_k(x_j; \theta^f) = \frac{1}{k} \sum_{i=1}^{k} f(x_j^{(i)}; \theta^f) \,, \tag{7}$$

where $x_j^{(i)}$ is the $i$th state visited by the Markov chain started at $x_j$. Since the Markov chain has a unique steady state distribution, $\bar{\bar{\hat{f}}}_k(x_j; \theta^f)$ is both asymptotically unbiased and consistent. That is $\lim_{k \to \infty} \bar{\bar{\hat{f}}}_k(x_j; \theta^f) \overset{a.s.}{=} \bar{\bar{f}}(x_j; \theta^f)$.

## 5 RESULTS

We evaluate our proposed augmentation procedure on multiple different tasks from the ATOM3D dataset (Townshend et al., 2020). Results are provided in Table 1.

### TASKS ON ATOM3D DATASETS

ATOM3D (Townshend et al., 2020) is a unified collection of datasets concerning the 3D structure of proteins. These datasets are specifically designed to provide a benchmark for ML methods which operate on 3D molecular structure, and represent a variety of important structural, functional, and engineering tasks. As part of our evaluation, we perform the (a) Protein Structure Ranking (PSR) (b) Ligand Efficacy Prediction (LEP) (c) Protein Mutation Stability Prediction (MSP) (d) Ligand Binding Affinity (LBA). We describe each of the tasks in detail in Appendix A.7. For all the datasets and tasks we report the same metrics as proposed by the authors in Townshend et al. (2020).

### MODEL, BASELINES AND DISCUSSION

We endow the vector gated GVP-GNN model (Jing et al., 2021), a GNN using GCN (Kipf & Welling, 2016) layers and the E(n) GNN Satorras et al. (2021) with conditional transformations from our proposed MCMC method. It is important to note that when the positions of the atoms are altered the protein graph which are inputs to the base encoders are changed appropriately in every epoch. As an ablation study, we also provide a strategy where all the transformations are created from "gold standard" $X_p$ rather than via the MCMC method, i.e., the MCMC is restarted at every epoch during training in the Appendix.

Looking at Table 1, we note that models augmented with our proposed conformer invariance framework, in general, outperforms the baseline models in multiple tasks for which conformer invariance is a requirement. In the LEP task, the proposed addition outperforms the baseline models for the E(n) GNN and the GVP-GNN, but not for the GNN(GCN). The LEP result for the GNN(GCN) is an oddity here, where a GNN which doesn't explicitly incorporate any atomic positional information outperforms all other models with information about atomic coordinates.

Additional results and ablation studies are presented in Appendix A.9.

### COMPUTATIONAL COMPLEXITY & SCALABILITY:

The directed forest for every protein is computed only once as a preprocessing step (a fixed ≈ 5-10 minutes per dataset). At every epoch, for a given protein, we select only one among all its constituent amino acids & perform a sequence of $3 \times 3$ matrix multiplications to obtain a new conformer. If $k$ denotes the number of atoms in the side chain of an amino acid, and height of a directed tree is in the order of $\log(k)$, the computational complexity to obtain a conformation is $\mathbf{O}(k \log k)$ (where $k \approx 15$

*in avg in our datasets)*, & this can be run in parallel for all proteins in a mini-batch. Molprobity (run in parallel again, currently runs externally on a web server via a command line call) adds 2s delay per minibatch (we could reduce to milliseconds if we integrated Molprobity into our code). The rejection rate from Molprobity is also very small (less than 1 reject in average across 100 molprobity calls). As an example, training (100 epochs) GVP-GNN - MSP requires $\approx 45$ min, and LBA requires $\approx 34$ min. In both cases our method requires an additional 4 min overhead. A complete training time table (on a Nvidia Tesla V100 GPU) for all models & datasets can be found in Appendix A.11.

## 6 RELATED WORK

Here, we present a high level summary of works related to ours. A more detailed comparison to the works mentioned below and others) are presented in Appendix A.6.

**Group Equivariant Neural Networks**: Group equivariant and invariant neural networks (Cohen & Welling, 2016; Lenssen et al., 2018; Kondor & Trivedi, 2018; Finzi et al., 2020; Hutchinson et al., 2021; Fuchs et al., 2020; 2021; Dehmamy et al., 2020) help capture discrete and continuous symmetries of elements (e.g. images) by introducing group theoretic constraints as inductive biases in the neural network. While group equivariant neural network capture global symmetries, local symmetries of manifold spaces can be captured via gauge equivariant networks (Cohen et al., 2019; De Haan et al., 2020). (Lenssen et al., 2018; Gerken et al., 2021; Bronstein et al., 2021) provide a complete review of the theoretical aspects and applications of group equivariant neural networks.

**Graph Neural Networks:** Graph Neural Networks (Kipf & Welling, 2016; Hamilton et al., 2017; Battaglia et al., 2018; Xu et al., 2018) have gained renewed focus over the past few years and have found applications in recommender systems, biology, chemistry, and many other real world problems which can be formulated as graphs and currently serve as the state of the art in majority of node and graph classification/ regression tasks. Graph neural networks work on the principles of permutation equivariance/ invariances (also groups) and have exploited a message passing framework to learn powerful and expressive representations of nodes/ graphs.

**Group Equivariant Graph Neural Networks:** These networks combine continuous symmetries (lie groups) with permutation equivariances and has found applications with resounding success on small molecules (Anderson et al., 2019; Klicpera et al., 2020; Satorras et al., 2021; Batzner et al., 2021) which exhibit rigid body characteristics. Employing (Farina & Slade, 2021), would make the neural network excessively invariant and allow the protein to be more flexible (allows unviable conformations) than it truly is. More recently, they have also been applied to learning representations of proteins, which is discussed below.

**Monte Carlo and MCMC Methods for Sampling Protein Conformations:** There has been a lot of prior work – (Boomsma et al., 2013; Olsson et al., 2013; Antonov et al., 2016; Irbäck & Mohanty, 2006; Vitalis & Pappu, 2009) which sample protein conformations by either defining/inheriting a distribution over the conformations (majorly based on the properties of the bonds, etc.) These methods can seamlessly be plugged into our framework as long as the MCMC is ergodic.

**Neural Networks for Representation Learning of Proteins:** Protein representation has gained a lot of attention especially with the tremendous successes of Alphafold and Alphafold2. A variety of neural network architectures including 3D CNNs, LSTM's and Transformers (treating the protein as a sequence) as well as graph neural networks have been employed to exploit the rigid body symmetries of proteins (Karimi et al., 2019; Pagès et al., 2019; Ingraham et al., 2019; Strokach et al., 2020; Baldassarre et al., 2021; Hermosilla et al., 2021; Jing et al., 2020; 2021).

**Generative Models:** Conformation generation models (Mansimov et al., 2019; Simm et al., 2020; Ganea et al., 2021; Xu et al., 2021b;a; Shi et al., 2021; Luo et al., 2021) have also gained attention recently with the goal to predict 3d structure of molecules given their 2d structure - our objective in this work is very different, but can be used to improve predictions of the aforementioned models.

## 7 CONCLUSIONS

This work addresses the limitations of current protein representation learning methods which are unable to learn conformer invariant representations — and hence unable to capture the inherent flexibility present in protein side chains pertinent to many downstream tasks. To address these, we introduced conditional transformations to capture protein structure, while respecting the restrictions posed by constraints on dihedral (torsion) angles and steric repulsions between atoms. Subsequently, we introduced a Markov chain Monte Carlo based framework to learn representations that are invariant to these conditional transformations.

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

# A  APPENDIX

## A.1  COMMONLY USED ACRONYMS IN THE PAPER

1. $G$ - group
2. $S_x$ - Set of transformations unique to element $x$
3. $t_x$ - Transformation action which transforms the element $x$
4. $m$ - Number of atoms in the protein under consideration
5. $n$ - Number of amino acids in the protein under consideration
6. $\boldsymbol{X}_s$ - Matrix of Scalar features associated with atoms in the protein $p$
7. $\boldsymbol{X}_p$ - Matrix of atomic coordinates associated with atoms in the protein $p$
8. $C_p$ - Set of conformations of protein $p$
9. $SO(3)$ - Group of all rotations in 3D space

## A.2  GROUP THEORY PRELIMINARIES

**Definition A.1** (Group). A group is a set $G$ equipped with a binary operation $\cdot : G \times G \to G$ obeying the following axioms:

- for all $g_1, g_2 \in G$, $g_1 \cdot g_2 \in G$ (closure).

- for all $g_1, g_2, g_3 \in G$, $g_1 \cdot (g_2 \cdot g_3) = (g_1 \cdot g_2) \cdot g_3$ (associativity).

- there is a unique $e \in G$ such that $e \cdot g = g \cdot e = g$ for all $g \in G$ (identity).

- for all $g \in G$ there exists $g^{-1} \in G$ such that $g \cdot g^{-1} = g^{-1} \cdot g = e$ (inverse).

**Definition A.2** (Group invariant functions). Let $G$ be a group acting on vector space $V$. We say that a function $f : V \to \mathbb{R}$ is $G$-invariant if $f(g \cdot x) = f(x) \ \forall x \in V, g \in G$.

**Definition A.3** ((Left) Group Action). For a group $G$ with identity element $e$, and $X$ is a set, a (left) group action $\alpha$ of $G$ on $X$ is a function $\alpha : G \times X \to X$ that satisfies the following two conditions:

1. Identity: $\alpha(e, x) = x, \forall x \in X$

2. Compatibility: $\alpha(g, \alpha(h, x)) = \alpha(gh, x)$

We will use a short hand of $g \cdot x$ for $\alpha(g, x)$ when the action being considered is clear from context.

## A.3  EXAMPLE DEMONSTRATING NON GROUP STRUCTURE OF A SET OF PROTEIN CONFORMATIONS

Consider $X$ to be a protein with n atoms and m amino acids and let the set of viable conformations of $X$ as $\{X_1, X_2, \ldots X_i, \ldots X_p\}$. Let $X_1$ be the conformation available in our dataset.

In each step of the transition, only a few atoms $(<< n)$ (atoms in a single amino acid of the entire protein- where the protein is made up of multiple hundreds of amino acids (m) in general) are subjected to an action from the SO(3) group here. The positions of all other atoms (outside that amino acid) in the side chain remain unaltered. So, in the $\mathbb{R}^{n \times 3 \times 3}$ matrix – most of the 3x3 entries are the identity matrix of 3 dimension.

Let $T_1^i \in \mathbb{R}^{n \times 3 \times 3}$ where $i \in \{1, 2, \ldots, p\}$ be the transformation which yields conformation $X_i$ from $X_1$.

Now consider $T_1^2$ (subscript of 1 since we start conformation $X_1$) which takes $X_1$ to $X_2$ and $T_1^3$ which takes $X_1$ to $X_3$ where $T_1^2$ and $T_1^3$, do not act on the same amino acid in the protein. Now, however, consider the case where performing $T_1^2$ and $T_1^3$ (or the other way around) sequentially would result in a case where atoms in two different amino acids would overlap (or come too

close to each other causing steric repulsion) - therefore resulting in a non viable conformation i.e. a composition of $T_1^2$ and $T_1^3$ acting on $X_1$, would not be present in the set of all allowed transformations $T_1 = \{T_1^1, \ldots T_1^i, \ldots T_1^p\}$. Therefore the set $T_1$ is not closed and doesn't form a group.

Also for two different conformations $X_1$ and $X_2$, their allowed transformation sets $T_1$, $T_2$ will not be identical (in the above example $T_1^3 \notin T_2$). It also easy to construct a case where $\bigcup_{i=1}^p T_i$ is not a group and all actions from this group acting on any given $X_i$ will not necessarily result in viable/ valid conformation.

### A.4 FLEXIBILITY ALLOWED BY OUR PROPOSED MODEL

Our proposed model allows every node in the directed tree to be rotated about its parents. For example, for the side chain shown in Figure 1 (Main Paper), the allowed flexibility is Figure 3.

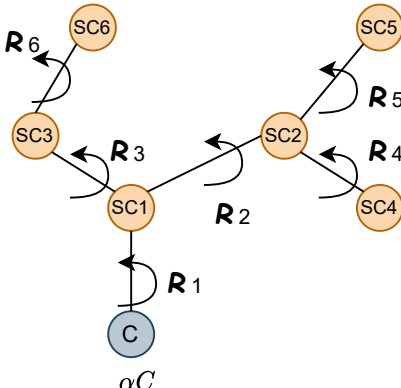

Figure 3: Maximum flexibility allowed by our candidate sampling process when the group associated with every node in the tree is SO(3) (the special orthogonal group in 3 dimensions). While not candidate is likely to be accepted such a candidate generation process provides the flexibility for every node to be rotated about its immediate parent while preserving bond lengths.

### A.5 PROOFS OF PROPOSITIONS

First, we restate and prove Proposition 3.3.

**Proposition A.4.** *Given the CSMC $\Phi_p$ from Definition 3.2 whose transitions are governed by $\kappa$ which is implicitly defined by Algorithm 1 as described above. For any pair of conformers $c_p, c'_p \in C_p$, there exists $\tau_p < \infty$, independent of $c_p$, such that $P_{\Phi_p}^{\tau_p}(c_p, c'_p) > 0$, where $P_{\Phi_p}^{\tau_p}$ is the $\tau_p$ step transition probability.*

*Proof.* Proof by construction. We prove the proposition by showing that one can construct a path $(c_p^{(1)} = c_p, \ldots, c_p^{(t)} = c'_p)$ such that $c_p^{(i)} \in C_p$ and $\kappa(c_p^{(i+1)}|c_p^{(i)}) > 0$, for all $0 < i < t$, and $t \leq T_p$. The trivial case where $c_p \equiv c'_p$ is proved since every group contains the identity element — sampling the identity element for every node in the directed tree yields the same conformation. Since we consider only non backbone transforming conformations, for the non trivial case, a maximum of $m - 4n$ atoms can differ in positions between any two conformations - where $m$ is the number of atoms in the protein and $n$ is the the number of amino acids in the protein $n$. Both $m, n$ are finite and we are dealing with continuous conformers (and continuous group actions about every node — groups are closed under their associated binary action, and SO(3) is path connected). So we can traverse between conformers (until we reach the desired conformer) sequentially in a finite number of steps, by using the constructed directed forest - selecting a amino acid (which doesn't violate the viability), fixing the positions of all other amino acids in the protein and rotating the side chain atoms in a single conformer to the final desired state. While this process may result in some side chains

being visited multiple times (due to viability constraints), considering continuous conformers and the SO(3) group (which is path connected) ensures we will never reach a state of deadlock. The second condition is satisfied because every group action has an inverse and we only use transformations from SO(3) for every node in the directed trees. □

Next, we restate and and prove Proposition 3.4

**Proposition A.5.** *The CSMC $\Phi_p$ defined in Definition 3.2 is uniformly ergodic if Proposition 3.3 is satisfied. Specifically there exists a unique steady state distribution $\pi_p$ such that for all $c_p \in C_p$, $\|P_{\Phi_p}^n(c_p, \cdot) - \pi_p(\cdot)\| \leq \mathtt{C} \, \mathtt{R}^n$, where $\mathtt{C} < \infty$ and $\mathtt{R} < 1$ are constants that depend on $\Phi_p$, $P_{\Phi_p}^n$ is the $n$ step transition probability and $\| \cdot \|$ is the $\ell_1$ norm.*

*Proof.* By Proposition 3.3, $\Phi_p$ satisfies Doeblin's condition as defined in page 396 of Meyn & Tweedie (2012) which states that for $c_p, c_p' \in C_p$, $P_{\Phi_p}^{T_p}(c_p, c_p') > \epsilon$ for some $\epsilon > 0$ [1]. The uniform ergodicity then holds due to Theorems 16.2.3 and 16.2.1 from Meyn & Tweedie (2012). □

Next, we restate and prove Proposition 4.2. We also restate the required assumptions for the proposition.

**Assumption A.6.** We make the following assumptions:

1. For any $\theta \in \Theta$ and $x_j^k \in S_j$, the function $f$ is differentiable $\forall j$

2. $\sup_{\theta \in \Theta, x_j^k \in S_j}\{\|\nabla_\theta \, \rho \circ f(x_j^k)\|\} < +\infty$ i.e. the gradients are bounded.

3. $\forall x_j^k \in S_j, \forall \theta_1, \theta_2 \in \Theta, \|\nabla_{\theta_1} \, \rho \circ f(x_j^k) - \nabla_{\theta_2} \, \rho \circ f(x_j^k)\| < L\|\theta_1 - \theta_2\|$ for some $L \geq 0$ i.e., the gradients are $L-$Lipschitz.

4. $\mathbb{E}_{x_j^k \sim \pi_j}[\nabla_\theta \, \rho \circ f(x_j^k)] = \nabla_\theta \, \mathbb{E}_{x_j^k \sim \pi_j}[\rho \circ f(x_j^k)]$

**Proposition A.7.** *Let the step sizes satisfy (5) and the function parameters $\theta$ be updated as (6) and Assumption 4.1 hold, then the MCGD optimization enjoys properties of almost sure convergence to the optimal $\theta$.*

*Proof.* Given that each protein has an associated time homogeneous Markov Chain with a unique steady state, independent of other proteins, the set of proteins in a mini-batch also form a Markov chain with a unique steady state. We then leverage Corollary 2 (Page 12) of Sun et al. (2018) along with Proposition 4.2 to ensure almost sure convergence to the optimal $\theta$. □

## A.6 EXTENDED RELATED WORK

Here, we elaborate on the related works section in Section 6

**Group Equivariant and Invariant Neural Networks**: Group equivariant and invariant neural networks (Cohen & Welling, 2016; Lenssen et al., 2018; Kondor & Trivedi, 2018; Finzi et al., 2020; Hutchinson et al., 2021; Fuchs et al., 2020; 2021; Dehmamy et al., 2020) help capture discrete and continuous groups symmetries of elements (e.g. images, point clouds). In this work, we learn representations which are invariant to symmetries which are not just groups, but to input dependent sets of transformations. To the best of our knowledge, our work is the first to consider conditional (input dependent) invariances.
Prior works on learning invariant models have leveraged Monte Carlo procedures (Finzi et al., 2020; Murphy et al., 2019b) to learn to be invariant to transformations of the input. Alternatively, our work constructs a Markov Chain with a unique steady state and leverages MCGD (Sun et al., 2018) to make it computationally tractable. Secondly, the aforementioned approaches are limited to being invariant to transformations from any specified Lie group with a surjective exponential map/ permutation group, while our work is not limited to groups. Thirdly, our theory ensures that every example/ object in the dataset can have a different set of input dependent, conditional transformations

---

[1]We note that this is a simplified version of the actual statement which is defined on the $\sigma$-algebra over $C_p$ denoted by $\sigma(C_p)$. Our proof holds when $c_p' \in \sigma(C_p)$

- and in fact can be seen as a generalization of the above works. In fact, the ablation study that we perform (Results provided in Appendix A.9 - Table 4) uses a Monte Carlo estimator and our MCMC procedure yields better performance than the Monte Carlo estimator.

While group equivariant neural network capture global symmetries, local symmetries of manifold spaces can be captured via gauge equivariant networks (Cohen et al., 2019; De Haan et al., 2020). Gauge symmetries require the manifold to be smooth – which is not the case for proteins. Moreover, different proteins have different sets of viable conformations, which would not be able to captured by standard gauge equivariant neural networks.

(Lenssen et al., 2018; Gerken et al., 2021; Bronstein et al., 2021) provide a complete review of the theoretical aspects and a wide variety of applications of group equivariant neural networks. A more comprehensive theoretical analysis of input dependent conditionally invariant neural networks is planned for future work.

**Graph Neural Networks:** Graph Neural Networks (GNNs) (Kipf & Welling, 2016; Hamilton et al., 2017; Battaglia et al., 2018; Xu et al., 2018) have gained renewed focus over the past few years and have found applications in recommender systems, biology, chemistry, and many other real world problems which can be formulated as graphs and currently serve as the state of the art in majority of node and graph classification/ regression tasks. Graph neural networks work on the principles of permutation equivariance/ invariances (Murphy et al., 2019a) (also groups) and have exploited a message passing framework to learn powerful and expressive representations of nodes/ graphs.

Both molecular graphs (both small molecules and macromolecules) as well as graphs based on intra molecular distances have been used with GNNs, to achieve state of the art for many molecular datasets and tasks (Hu et al., 2020; Morris et al., 2020). Here, we leverage three graph based neural networks as baselines for our model.

**Group Equivariant Graph Neural Networks:** Group Equivariant GNNs combine continuous symmetries (lie groups such as SE(3), E(3), SO(3)) with permutation equivariances and has found applications with resounding success on small and large molecules (Anderson et al., 2019; Klicpera et al., 2020; Satorras et al., 2021; Batzner et al., 2021).

However, the methods are only able to capture rigid body characteristics of molecules and while capturing the above lie group symmetries is also able to capture input depedent transformations. Employing (Farina & Slade, 2021), would make the neural network excessively invariant and allow the protein to be more flexible (allows unviable conformations) than it truly is. More recently, they have also been applied to learning representations of proteins, which is discussed below.

**Monte Carlo and MCMC Methods for Sampling Protein Conformations:** There has been a lot of prior work – (Boomsma et al., 2013; Olsson et al., 2013; Antonov et al., 2016; Irbäck & Mohanty, 2006; Vitalis & Pappu, 2009) which sample protein conformations by internal coordinate transformations. However, The goals of the existing MCMC methods are significantly different compared to ours. The existing methods define/inherit a distribution over the conformations (majorly based on the properties of the bonds, etc.) and then aim to sample highly probable conformations from this distribution. Our method, is much simpler and only requires that the chain being used is ergodic and is invariant to the actual form of the distribution. We note that the existing Markov chains can seamlessly be used as drop-in replacements to sample conformations as part of our framework as long as the MCMC is ergodic. We consider studying the impact of different MCMC methods (which sample from different distributions) and their influence on the performance in different tasks as important future work.

**Neural Networks for Representation Learning of Proteins:** Protein representation has gained a lot of attention especially with the tremendous successes of Alphafold and Alphafold2. A variety of neural network architectures including 3D CNNs, LSTM's and Transformers (treating the protein as a sequence) as well as graph neural networks have been employed to exploit the rigid body symmetries of proteins (Karimi et al., 2019; Pagès et al., 2019; Ingraham et al., 2019; Strokach et al., 2020; Baldassarre et al., 2021; Hermosilla et al., 2021; Jing et al., 2020; 2021).

In this work, while we use GNNs and Group Invariant GNNs as a part of the model, we note that we can equally replace them with CNNs, LSTMs, Transformers and other models used for proteins without any change in the underlying theory.

**Tree Construction Method for Molecules**: Jin et al. (2018), in their work JTVAE, propose

- a procedure to construct tree for molecules. We note that JTVAE is a more general purpose approach for constructing trees from molecules and may not be the best suited for proteins where the backbone atoms are largely observed (experimentally) to be rigid.

**Generative Models:** Protein conformation generation models (Mansimov et al., 2019; Simm et al., 2020; Ganea et al., 2021; Xu et al., 2021b;a; Shi et al., 2021; Luo et al., 2021) have also recently gained attend where the goal of the model is to predict 3d structure of molecules given input 2d structure - our objective in this work is completely different, but can be used to improve predictions of the aforementioned models.
While our model is explicitly not a generative model, our framework can be leveraged towards generative modeling with the help of tools such as noise outsourcing (Chapter 6) (Kallenberg, 2006) and we see this as important future work.

**Non-Rigid Body Dynamics**: Non-Rigid Body Dynamics of objects has long been studied both by physicists and in the fields of computer vision to understand and capture the geometric deformations of objects (Taylor et al., 2010; Masci et al., 2015). To the best of our knowledge, there exists no prior work in deep learning which captures the non rigidity of protein molecules (which cannot be modeled as $C^k$ manifolds). As important future work, we would like to study the impact of leveraging input dependent conditional invariances for modeling other geometric objects (which are $C^k$ manifolds) as well as images and robotics (e.g. the symmetries for a humanoid is different from that of a tractor).

**Unrelated Work with similar names:** Non classical and conditional symmetries of solutions to ODE's, PDE's have been discussed in the past - these works while they share a similar title, have very little in common as we are not dealing with jet spaces or manifolds (Joseph, 1968; Fushchich & Zhdanov, 1992; Olver & Vorob'ev, 1996).

## A.7 DETAILS ABOUT DATASETS AND TASKS

In this section, we describe briefly each of the datasets (and their associated tasks). Information about the splits and license information is provided in Table 2.

**PSR:** This task utilizes data from the structural models submitted to the Critical Assessment of Structure Prediction competition (CASP - Kryshtafovych et al. (2019) - a blind protein structure prediction competition) to rank protein structures from the experimentally determined structure of the protein. The problem is formulated as a regression task, where we predict the global distance test of each structural model from the experimentally determined structure. As prescribed by the dataset authors, the dataset is split by competition years.

**MSP:** The goal of this task is to identify mutations that stabilize a protein's interactions which forms an important step towards the design of new proteins. This task is significant as probing mutations experimentally techniques are labor-intensive. Atom3D (Townshend et al., 2020) derives this dataset by collecting single-point mutations from the SKEMPI database (Jankauskaitė et al., 2019) and model each mutation into the structure to produce a mutated structure. The learning problem is then formulated as a binary classification task where the goal is to predict whether the stability of the complex increases as a result of the mutation. We employ the same splits as suggested by the dataset authors wherein the protein complexes are split such that no protein in the test dataset has more than 30% sequence identity with any protein in the training dataset.

**LBA:** This task deals with the problem of predicting the strength (affinity) of a candidate drug molecule's interaction with a target protein. The dataset is constructed using the PDBBind database (Wang et al., 2004; Liu et al., 2015), a curated database containing protein-ligand complexes from the PDB and their corresponding binding strengths (affinities). The task is formulated as a regression task with the goal to predict $pK = -\log_{10}(K)$, where $K$ is the binding affinity in Molar units. The splits are created such that no protein in the test dataset has more than 30% sequence identity with any protein in the training dataset.

**LEP:** The shape of protein impacts whether a protein is in an on or off state which plays an important role in predicting the shape a protein will favor during drug design.This dataset is obtained by curating proteins from several families with both "active" and "inactive" state structures, and model in 527 small molecules with known activating or inactivating function using the program Glide (Friesner

et al., 2004). The task is formulated as a binary classification task where the goal is to predict whether a molecule bound to the structures will be an activator of the protein's function or not. We use the same split as recommended by the ATOM3D authors.

Table 2: Summary of the datasets

| Task | # Train | # Val | # Test | Original Source | License |
|------|---------|-------|--------|-----------------|---------|
| MSP | 2864 | 937 | 247 | SKEMPI (Jankauskaitė et al., 2019) | Creative Commons CC-BY |
| LBA | 3563 | 448 | 452 | PDBBlind (Wang et al., 2004) | Creative Commons NonCommercial-NoDerivs (CC-BY-NC-ND) |
| LEP | 304 | 110 | 104 | PDB (Berman et al., 2000) | Creative Commons CC-BY |
| PSR | 25400 | 2800 | 16099 | CASP (Kryshtafovych et al., 2019) | Creative Commons CC-BY |

## A.8 EXPERIMENTAL SETUP

The code for the baseline models (GVP-GNN (Jing et al., 2021), E(N) GNN (Satorras et al., 2021) and GNN(GCN) (Townshend et al., 2020; Kipf & Welling, 2016)) were used as provided by the authors (licenses as dictated by the code authors). Our conformer invariance implementation is in PyTorch using Python 3.8. We also leverage networkx to create the directed forests. For all three models we tune the hyperparameters – learning rate ($\in \{0.1, 0.01, 0.001, 0.0001\}$) and mini batch size ($\in \{4, 8, 16, 32, 64\}$). For the E(n) GNN model - since there have been no previous models for the aforementioned protein tasks - we also tune the number of GNN layers ($\in \{4, 5, 6, 7\}$) as a hyper parameter. The experiments were all performed on Tesla V100 GPU's. For more details refer to the code provided.

## A.9 ADDITIONAL RESULTS

In Table 3, we present the results, including additional datasets and tasks, than presented in the main paper. In the LEP task, the proposed addition outperforms the baseline models for the E(n) GNN and the GVP-GNN, but not for the GNN(GCN). The LEP result for the GNN(GCN) is an oddity here, where a GNN which doesn't incorporate any rigid body transformations outperforms all other models.

In Table 4, we present an ablation study, where we provide a strategy where all the transformations are created from "gold standard" $X_p$ rather than via the MCMC method, i.e., the MCMC is restarted at every epoch during training. From the table, we note that the MCMC method tends to outperform the non MCMC method, which can be attributed to the guarantees it provides to the learning framework.

Table 3: GNN(GCN), GVP-GNN, E(N) GNN - Baseline vs Conformation Invariant Strategies for multiple different tasks on proteins from the ATOM3D dataset. Corresponding to the metric, ↑ indicates that higher is better, while ↓ indicates that lower is better. Bold values indicate best results for a given row. The values for GNN were obtained from Townshend et al. (2020) and for the GVP-GNN from Jing et al. (2021). Gray colored cells indicates that the augmented model outperforms the baseline model.

| Task | Metric | Baseline (GNN[GCN]) | MCMC Augmented GNN (**Ours**) | Baseline (GVP GNN) | MCMC Augmented GVP-GNN (**Ours**) | Baseline (E(N) GNN) | MCMC Augmented E(N) GNN (**Ours**) |
|------|--------|---------------------|------------------------------|--------------------|-----------------------------------|---------------------|-----------------------------------|
| PSR | Global $R_s$ ↑ | $0.755 \pm 0.004$ | $0.761 \pm 0.004$ | $0.845 \pm 0.004$ | $\mathbf{0.852 \pm 0.006}$ | $0.827 \pm 0.004$ | $\mathbf{0.852 \pm 0.004}$ |
| LEP | AUROC ↑ | $\mathbf{0.740 \pm 0.010}$ | $0.672 \pm 0.012$ | $0.628 \pm 0.055$ | $0.704 \pm 0.039$ | $0.677 \pm 0.014$ | $0.714 \pm 0.005$ |
| LBA | RMSE ↓ | $1.570 \pm 0.025$ | $1.519 \pm 0.022$ | $1.594 \pm 0.073$ | $1.435 \pm 0.007$ | $1.392 \pm 0.001$ | $\mathbf{1.384 \pm 0.011}$ |
| MSP | AUROC ↑ | $0.621 \pm 0.009$ | $0.662 \pm 0.008$ | $0.680 \pm 0.015$ | $\mathbf{0.857 \pm 0.049}$ | $0.652 \pm 0.006$ | $0.843 \pm 0.037$ |

## A.10 CASE AGAINST USING AVERAGE REPRESENTATIONS IN TRAINING AND FOR REQUIRING A SINGLE INVARIANT REPRESENTATION

The case for the requirement of a single conformer invariant representations – can be seen from the fact that different protein conformations, say $X_1, X_2$ of the same protein $X$, may be seen during

Table 4: GVP-GNN, GNN (GCN) - Baseline vs Ablation vs Conformation Invariant Strategies for four different tasks on proteins from the ATOM3D (Townshend et al., 2020) dataset. Corresponding to the metric, ↑ indicates that higher is better, while ↓ indicates that lower is better. Bold values indicate best results for a given row. Gray colored cells indicates that the augmented model outperforms the baseline model.

| Task | Metric | Baseline (GVP GNN) Jing et al. (2021) | Non MCMC Augmented GVP-GNN (Ablation) | MCMC Augmented GVP-GNN (Ours) | Baseline (GNN) Townshend et al. (2020) | Non MCMC Augmented GNN (Ablation) | MCMC Augmented GNN (Ours) |
|------|--------|------|------|------|------|------|------|
| PSR | Global $R_s$ ↑ | $0.845 \pm 0.004$ | $0.806 \pm 0.011$ | $\mathbf{0.852 \pm 0.006}$ | $0.755 \pm 0.004$ | $0.766 \pm 0.001$ | $0.761 \pm 0.004$ |
| LEP | AUROC ↑ | $0.628 \pm 0.055$ | $\mathbf{0.739 \pm 0.060}$ | $0.704 \pm 0.039$ | $\mathbf{0.740 \pm 0.010}$ | $0.657 \pm 0.008$ | $0.672 \pm 0.012$ |
| LBA | RMSE ↓ | $1.594 \pm 0.073$ | $1.635 \pm 0.007$ | $\mathbf{1.435 \pm 0.007}$ | $1.570 \pm 0.025$ | $1.520 \pm 0.022$ | $1.519 \pm 0.022$ |
| MSP | AUROC ↑ | $0.680 \pm 0.015$ | $0.799 \pm 0.016$ | $\mathbf{0.857 \pm 0.049}$ | $0.621 \pm 0.009$ | $0.610 \pm 0.021$ | $0.662 \pm 0.008$ |

train and test phases. Without conformer invariant representations - this may lead to different representations and therefore different predictions for the same protein (bad).

One may argue, that an approximate conformer invariant neural network can be learned by averaging representations of multiple Monte Carlo conformations during training. However, this is computationally expensive as this would need to back-propagate and update parameters for multiple conformations for every protein in every epoch – bad as this leads to an exponential overhead. On the other hand, our procedure with MCGD, in every epoch uses only one conformation from our Markov Chain and still ensures convergence to optimal parameters. It is important to note that, during inference we still do average representations over multiple conformations (from our Markov chain) to output conformer invariant representations - which again due to MCGD training procedure and the unique steady state of our Markov Chain, ensures confomer invariant representations are achieved – which yield better performance than current state of the art on multiple datasets and tasks. As stated in the main paper, our framework can work well with other Markov chains as well. We chose the proposed chain purely because it is simpler than the existing methods (in that it doesn't require a distribution over conformations to be assumed) and as such is easier to sample from.

### A.11 TRAINING TIME

In Table 5, we present the training time (for 100 epochs) for each of the baseline models as well as well for models with our proposed conformer invariance framework addition. From the table, we note that, on an average the increase in training time over 100 epochs is $\approx$ 3-4 minutes which is negligible in comparison to the training time (without the proposed framework addition).

Table 5: Training Time (in minutes for 100 epochs – rounded to closest integer) - GNN(GCN), GVP-GNN, E(N) GNN - Baseline vs Conformation Invariant Strategies for multiple different tasks on proteins from the ATOM3D dataset. Lower is better.

| Task | Baseline (GNN[GCN]) | MCMC Augmented GNN (Ours) | Baseline (GVP GNN) | MCMC Augmented GVP-GNN (Ours) | Baseline (E(N) GNN) | MCMC Augmented E(N) GNN (Ours) |
|------|------|------|------|------|------|------|
| PSR | 184 | 190 | 1112 | 1118 | 485 | 492 |
| LEP | 6 | 8 | 8 | 9 | 10 | 11 |
| LBA | 26 | 28 | 34 | 38 | 59 | 62 |
| MSP | 58 | 61 | 45 | 49 | 125 | 128 |

