# OpenReview forum: "Conditional Invariances for Conformer Invariant Protein Representations"
_ICLR.cc/2023/Conference — Submitted to ICLR 2023_

### Official Review · Reviewer_bwoG · 2022-10-23

**Confidence:** 4
**Correctness:** 4
**Technical Novelty And Significance:** 4
**Empirical Novelty And Significance:** 2
**Recommendation:** 3

**Clarity, Quality, Novelty And Reproducibility:**

- The paper is well-written and easy to follow.
- The approach to sample protein conformations is novel at the extent of the reviewer’s knowledge.
- The authors didn’t provide a code to reproduce their results.


**Strength And Weaknesses:**

Strength:
- The directed forest construction and conditional transformation of sub-structures provide a principled approach to sample conformations given a protein structure.
- The authors formally show that the MCGN optimization ensures the convergence of conformer invariant learning procedure.

Weakness:
- The approach assumes that the protein molecule can be decomposed into a directed forest structure. This assumption is often not true for realistic proteins. The authors use some arbitrary rules to deal with cycles/rings: “cycles/rings which are present in the molecule are broken on the basis of bond length, i.e., larger bonds are chosen before smaller bonds. Ties between same-length bonds are broken arbitrarily.”The resulting conformations are likely unrealistic due to these arbitrary rules.
- Consequently, the authors use a validity checker (Molprobity) to filter invalid conformations. It means that the sampled structures largely depend on the validity checker. It is unclear if the conditional invariant sampling approach provides benefits compared with a prior MCMC sampling approach. It will also be beneficial if the authors can report the acceptance rate of the MCMC sampling.
- The performance improvement over a GNN baseline is marginal (Table 1). It is unclear if the proposed approach can outperform a much simpler data argumentation baseline, e.g., random Gaussian noise + validity filtering.
- Have the authors considered leveraging substructures similar to the JTVAE paper by Jin et al. (ICML 2018) for the construction of the directed forest?


**Summary Of The Paper:**

This paper aims to learn a conditional invariant representation for protein structure by introducing a data augmentation approach. The approach augments the protein 3D conformations by applying conditional invariant transformations to sub-structures in protein with a novel Markov chain Monte Carlo (MCMC) sampling. The paper demonstrates improved performance over a GNN baseline that inputs the initial 3D structure of protein without data argumentation.

**Summary Of The Review:**

In summary, the authors introduce a novel conditional invariant approach to argument the protein conformations and improve the performance in property prediction tasks. Despite the nice theory, the performance improvement seems marginal and there are some realistic assumptions in the direction forest construction approach. I feel this work requires further improvement and more baselines to demonstrate its usefulness in realistic tasks.

---

> ### Author Response · Authors · 2022-11-19
> **Response to Reviewer bwoG (Part 1/2)**
>
> We would like to thank the reviewer for their review. We address their concerns below.
>
> **Q1**: The approach assumes that the protein molecule can be decomposed into a directed forest structure. This assumption is often not true for realistic proteins. The authors use some arbitrary rules to deal with cycles/rings: “cycles/rings which are present in the molecule are broken on the basis of bond length, i.e., larger bonds are chosen before smaller bonds. Ties between same-length bonds are broken arbitrarily.”The resulting conformations are likely unrealistic due to these arbitrary rules.
>
> **A1**: Thank you for bringing this up. We agree with the reviewer partly here that an arbitrary decomposition of a protein into directed forest may oftentimes not be realistic  - however we argue why this is not consequential in our case. We understand your concern that it may either lead to undesirable conformations or alternatively that not all the potential conformations, pertinent to a given protein may be explored. We would like to clarify that indeed all possible viable conformations (for a given side chain) can be reached using the directed forest construction and the MCMC procedure - regardless of the protein - as long as it is viable (which we check using Molprobity - because of ease of use - but it can be replaced with other alternatives - as we mention in the paper). We understand the confusion as there could be cases when a bond is broken during the tree construction and therefore not allowing rotation about the particular bond. However, it is important to note that the two atoms would share a common ancestor in the tree and given that the group of rotations SO(3) is path connected - it would allow for all possible viable conformations through a sequence of alternative actions (More details also in proof of proposition 3.3). Moreover this strategy, while more simplistic, is very similar to the underlying procedure followed in the widely used protein conformation sampler -- PROFASI, [Irback and Mohanty, 2006].
>
> **Q2**: Consequently, the authors use a validity checker (Molprobity) to filter invalid conformations. It means that the sampled structures largely depend on the validity checker. It is unclear if the conditional invariant sampling approach provides benefits compared with a prior MCMC sampling approach. It will also be beneficial if the authors can report the acceptance rate of the MCMC sampling.
>
> **A2:** To clarify, as we specify in the paper, the main goal of our proposal is to introduce a simplistic MCMC procedure (in comparison to prior works - without making explicit assumptions on the underlying distribution) to learn conformer invariant representations - and empirically demonstrate that even with a simple approach there are considerable performance gains without adding excessive computation time (average of 4 minutes additional total training time).. In this regard,  we would like to point out that just with this change, we observe up to 26 % improvement over the baselines.  Regarding the acceptance rate of the MCMC sampling with Molprobity - as we had mentioned in the computation complexity and scalability section of the main paper - The rejection rate from Molprobity is also very small (less than 1 reject in average across 100 molprobity calls).

---

> > ### Author Response · Authors · 2022-11-19
> > **Response to Reviewer bwoG (Part 2/2)**
> >
> > **Q3:** The performance improvement over a GNN baseline is marginal (Table 1). It is unclear if the proposed approach can outperform a much simpler data argumentation baseline, e.g., random Gaussian noise + validity filtering.
> >
> > **A3:** Thank you for bringing this up. The single case of performance of the GNN baselines on the LEP task is certainly an anomaly as observed even by the authors of ATOM 3D dataset - that a model which does not consider atomic positions performs better than models which do). We would like to emphasize that our proposed procedure performs worse than the baseline in only 1 out of the 12 scenarios (4 tasks x 3 baselines). With regard to other baselines, as a part of our preliminary studies, we did test the case where in every epoch we:
> > 1. Rotated about the just the bond between the $\alpha$-Carbon (backbone atom) and the $\beta$-Carbon of a single amino acid (Starting from the original conformation in the dataset in every epoch)
> > 2. Rotated about the bond between the $\alpha$-Carbon (backbone atom) and the $\beta$-Carbon of all amino acids in the protein. (Starting from the original conformation in the dataset in every epoch)
> > 3. Rotated about a random bond in the side chain in a single amino acid of the protein ((Starting from the original conformation in the dataset in every epoch). This is the ablation present in the appendix currently
> >
> > We observed worse results in 1 and 2 in comparison to 3 (current ablation study in appendix), but better than the baseline models. However  our proposed MCMC approach from the main paper outperforms 3. We are currently running extensive tests for all the datasets (with complete hyper-parameter search), and will update before Dec 12 (author - reviewer discussion deadline) with this ablation study.
> >
> > **Q4:** Have the authors considered leveraging substructures similar to the JTVAE paper by Jin et al. (ICML 2018) for the construction of the directed forest?
> >
> > **A4**: Thank you for pointing out an alternative tree construction procedure - we have added a reference to this in our extended related work section in the appendix - and this is a nice direction for potential future work. We would however also like to note that JTVAE is a more general purpose approach for constructing trees from molecules and may not be the best suited for proteins where the backbone atoms are largely observed (experimentally) to be rigid.
> >
> > Please let us know if you are satisfied with our responses and/ or have any further queries that we can address - and we will add our responses before the end of the Private Reviewer-Author Discussions deadline (12 December - per the Author Guide on ICLR portal). If we have addressed your concerns, please consider revising your score.
> >
> > **References:**
> > 1. Irbäck, Anders, and Sandipan Mohanty. "PROFASI: a Monte Carlo simulation package for protein folding and aggregation." Journal of computational chemistry 27.13 (2006): 1548-1555.
> > 2. Jin, Wengong, Regina Barzilay, and Tommi Jaakkola. "Junction tree variational autoencoder for molecular graph generation." International conference on machine learning. PMLR, 2018.

---

### Official Review · Reviewer_zqSH · 2022-10-23

**Confidence:** 4
**Correctness:** 2
**Technical Novelty And Significance:** 3
**Empirical Novelty And Significance:** 2
**Recommendation:** 6

**Clarity, Quality, Novelty And Reproducibility:**

The idea in the paper is original. No source code was included in the submission, it is hard to reproduce the results.
The paper was well written. However, it makes too many assumptions about the distribution of the conformation without clear support both theoretical and empirical evidence (see my comments on weaknesses). A large number of related works in protein representation learning using amino acid sequences were ignored.




**Strength And Weaknesses:**

Strength
This is an interesting and novel idea on conformation sampling.

Good experimental results compared to the baseline that does not leverage augmented data.

Weaknesses


I see a lot of strong assumptions without thorough validation and supported evidence both theoretically and empirically

Step 4 in Algorithm 1: trees were sampled independently. Is there any evidence that the conformation of side chains of different amino acids are independent of each other?

Representing an amino acid as a tree requires breaking circles (if exists) at a random point, different breaking points result in different tree structures. The current conformation sampling approach assumes conditional independence between a node and all of its grand-parent given the parent nodes. This assumption seems a strong assumption that may lead to ill-approximation of the true conformation distribution. Is there any way to validate that the distribution of the sampled data fits the right distribution of conformation?


Since evidence about the approximation of the true conformation distribution is not provided, to demonstrate that the proposed conformation sampling approach based on the atom forest is a more effective data augmentation approach than other simple data augmentation approaches, the following simple  data augmentation should be considered as a baseline to compare to:
+ For each back-bond atom, keep the back-bond atom fixed, and randomly transform all the side-chain nodes associated with that back-bond atom using a random rotation matrix.

This baseline will validate that the data augmentation process proposed in the paper is an effective data augmentation approach compared to this simple data augmentation baseline.


A lot of relevant existing works in protein representation using amino acid sequences are ignored. When protein 3D structure data is scarce because of expensive acquisition, a lot of work on protein representation learning relies on linear amino acid sequences. Although this representation of protein does not tell much about the 3D conformations of protein explicitly, the availability of a large database of an amino acid sequence is very relevant for self-supervised learning to be trained on. The discussion of these works in the related work and comparison to these works should be conducted carefully. I would suggest the authors compare to at least the following baseline representation trained on large amino acid sequence databases, https://github.com/facebookresearch/esm:
+ ESM-1b
+ ESM-MSA-1b


**Summary Of The Paper:**

Summary of the work
The paper studies protein representation learning from the 3D structure. Recent works have shown that although the back-bond of protein 3D structure is fixed, the side chains associated with each back-bone atom are flexible and may correspond to different conformations.

The authors proposed a method to augment the current 3D structures with randomly sampled conformations from the gold 3D structure. The samples are used as additional data for both training and inference in downstream learning tasks where the representation of the proteins is approximated by the average of the representation of the conformations.

They show that this simple data augmentation approach improves the performance of supervised downstream tasks that only relies on 3D CNN. 3D GCN GNN to transform the gold conformations in the Atomic3D benchmark dataset.

Even though methods for sampling confirmation using MCMC exist in the literature, a novelty of the work relies on the new sampling approach that constructs for each back-bond atom a directed tree by breaking the loops in the side chains associated with the back-bond atom. Sampling was done by the sampling point set of the nodes in the trees starting from the root nodes down to the leaves, where the later point set is conditionally dependent on the parent's conformations. An MCMC approach is used to sample conformations of the side chains from these point sets.




**Summary Of The Review:**


I see this work as interesting and may have a good impact on the field. But some points need to improve especially stronger support for the proposed conformation sampling approach. I proposed additional experiments that the authors should consider to strengthen the support evidence, I will be happy to change my score once these additional experiments are added to the papers and the results are still significant concerning the new baselines.  see this work interesting and may have a good impact in the field. But there are points that need to improve especially a stronger support for the proposed conformation sampling approach. I proposed additional experiments that the authors should consider to strengthen the support evidence, I will be happy to change my score once these additional experiments are added to the papers and the results are still significant with respect to the new baselines.

---

> ### Author Response · Authors · 2022-11-19
> **Response to Reviewer zqSH (Part 1/2)**
>
> We would like to thank the reviewer for their review. We address their concerns below.
>
> **Q1:** Step 4 in Algorithm 1: trees were sampled independently. Is there any evidence that the conformation of side chains of different amino acids are independent of each other?
>
> **A1**: Thank you for raising this concern. No, we are not aware of any experimental evidence which showcases this. However, we understand your concern could be because of the potential issue that not all conformations, pertinent to a given side chain may be explored if it is broken differently to the same amino acid in a different protein. We would like to clarify that indeed all possible conformations (for a given side chain) can be reached using the directed forest construction and the MCMC procedure - regardless of the protein - as long as it is viable. We understand the confusion as there could be cases when a bond is broken during the tree construction and therefore not allowing rotation about the particular bond. However, it is important to note that the two atoms would share a common ancestor in the tree and given that the group of rotations SO(3) is path connected - it would allow for all possible viable conformations through a sequence of alternative actions (We provide more in the proof of proposition 3.3). Given that the ultimate goal in our case is conformer invariance, and that we can reach all possible conformations, the independence assumption doesn’t affect the final desiderata.
>
>
> **Q2:** Representing an amino acid as a tree requires breaking circles (if exists) at a random point, different breaking points result in different tree structures. The current conformation sampling approach assumes conditional independence between a node and all of its grand-parent given the parent nodes. This assumption seems a strong assumption that may lead to ill-approximation of the true conformation distribution. Is there any way to validate that the distribution of the sampled data fits the right distribution of conformation?
>
> **A2:** We would like to clarify that the assumption made above is not true in its entirety. The position of a node still needs to preserve the sum of bond distances to the grandparent while still preserving its viability. As we stated in the previous answer -  all possible conformations (for a given side chain) can be reached using the directed forest construction procedure and the MCMC procedure. Moreover this strategy, while more simplistic, is very similar to the underlying procedure followed in the widely used protein conformation sampler -- PROFASI, [Irback and Mohanty, 2006].
>
> **Q3:** Since evidence about the approximation of the true conformation distribution is not provided, to demonstrate that the proposed conformation sampling approach based on the atom forest is a more effective data augmentation approach than other simple data augmentation approaches.
>
> **A3:** Thank you for this suggestion. As a part of our preliminary studies, we did test the case where in every epoch we:
>
> 1. Rotated just about the bond between the $\alpha$-Carbon (backbone atom) and the $\beta$-Carbon of a single amino acid (Starting from the original conformation in the dataset in every epoch)
> 2. Rotated about the bond between the $\alpha$-Carbon (backbone atom) and the $\beta$-Carbon of all amino acids in the protein. (Starting from the original conformation in the dataset in every epoch)
> 3. Rotated about a random bond in the side chain in a single amino acid of the protein ((Starting from the original conformation in the dataset in every epoch). This is the ablation present in the appendix currently.
>
> We observed worse results in 1 and 2 in comparison to 3 (current ablation study in appendix), but better than the baseline models. However  our proposed MCMC approach from the main paper outperforms 3. We are currently running extensive tests for all the datasets (with complete hyper-parameter search), and will update before Dec 12 (author - reviewer discussion deadline) with this ablation study.
>
> **Q4**: A lot of relevant existing works in protein representation using amino acid sequences are ignored.
>
> **A4:** Thank you for bringing this up. We would like to emphasize that we have solely focused on the ternary structure of proteins and consider the amino acid sequence based formulation as beyond scope. However, we will add the contemporary work (ESM on proteins - code released Aug 22) as well as other amino acid sequence based modeling works into the extended related section in the appendix.

---

> > ### Author Response · Authors · 2022-11-19
> > **Response to Reviewer zqSH (Part 2/2)**
> >
> > Please let us know if you are satisfied with our responses and/ or have any further queries that we can address - and we will add our responses before the end of the Private Reviewer-Author Discussions deadline (12 December - per the Author Guide on ICLR portal). If we have addressed your concerns, please consider revising your score.
> >
> > **References**:
> > 1. Irbäck, Anders, and Sandipan Mohanty. "PROFASI: a Monte Carlo simulation package for protein folding and aggregation." Journal of computational chemistry 27.13 (2006): 1548-1555.
> > 2. Lin, Zeming, et al. "Language models of protein sequences at the scale of evolution enable accurate structure prediction." bioRxiv (2022).

---

### Official Review · Reviewer_W5rQ · 2022-10-24

**Confidence:** 4
**Correctness:** 3
**Technical Novelty And Significance:** 3
**Empirical Novelty And Significance:** Not applicable
**Recommendation:** 5

**Clarity, Quality, Novelty And Reproducibility:**

## Clarity
The paper is well written with clear figures to support the story.

## Quality
The manuscript seems technically sound. The method is supported with proofs in the appendix.

The result section should include a baseline where the MCMC is done offline, rather than as part of the training procedure. If a pool of samples could be generated prior to training and used as standard data augmentation (e.g. sampling randomly from the pool in each epoch), it would simplify the training procedure considerably. The Additional Result section has a "gold standard" baseline, which might actually be the baseline I request, but is not very clearly explained. The authors should describe how the "gold standard" is created - how long the offline MCMC has been run to create the gold standard (is it fully converged?), and what they mean when they write that the "MCMC is restarted in every epoch during training". If the method is indeed better than the offline method, it would be helpful if the authors could describe explicitly why they think that their approach is better than simple data augmentation (currently, it says "which can be attributed to the guarantees it provides to the learning framework" - which was not clear to me. Which guarantees are these?)

## Reproducibility
As far as I could see, no code was shared as part of this submission, and there was also no Code Availability statement in the manuscript. The authors should state if they intend to share the code.


## Minor comments

Caption, Figure 1. "Just the alpha carbon and the side chain atom". "atom" -> "atoms"

"Traditionally, protein structures are solved by X-ray crystallography or cyro-EM". "cyro-EM" -> "cryo-EM"

"and the obtained structure is normally considered a unique 3D conformation of the molecule."
It is unclear what "unique" means here". Do you mean "representative"?

Proposition 3.3 and Proposition 3.4. The authors should state if these results are any different from the general results known for MCMC simulations of proteins. I would think that it is well known that transitions kernels such as the one described would lead to a Markov chain with a unique steady state distribution. If this is indeed the case, the authors should state that they are simply restating known results. If not, they should state how their result is different.

"Then, a simple way to obtain conformer invariant representations of protein x_j (apart from using trivial functions such as a constant function or function independent of X_v)"
What is X_v here?


**Strength And Weaknesses:**

The paper does a good job of formally describing the problem and the details of the proposed algorithm. The manuscript seems technically sound.  However, I struggle with the motivation for the method. The authors introduce the conditional invariances to sidechain conformations as a fundamental property on par with rotational and translation invariances of the global structure of the molecule, but this equivalence seems shaky. The standard desire for invariance to rotation and translation of a molecular structure has a clear motivation: we wish any result to be independent on the arbitrarily chosen coordinate system of our 3D coordinate system. Or in other words, the protein structure that we care about has 6 degrees of freedom fewer than when we parameterize it using Cartesian coordinates of all its atoms. From a physical perspective, if we are considering intrinsic properties of a protein, for instance its stability, the position and rotation of a protein is fundamentally irrelevant. All physical interactions between atoms in the molecule are independent of the global orientation and translation.

In contrast, the invariance towards sidechain conformations is less clear cut. The authors state that "for most proteins, regardless of their side chain conformation (as long as viable) under consideration – their protein fold class/ other scalar properties remain the same, their mutation (in)stability remains unaltered, protein ligand binding afﬁnity (apart from changes at the ligand binding site) remain the same, etc.". While it is true that the fold class stays the same because it is typically defined only based on the backbone conformation of a protein, the remaining properties (stability, binding affinity), are certainly dependent on side-chain conformations. The keyword here seems to be "as long as viable", which implicitly establishes the equivalence of different structures. But what is deemed "viable" will depend on the physical quantity we wish to predict, and the resolution at which we are modelling the system. The desired level of "invariance" will thus depend on the modelling task. For some tasks, the exact sidechain orientations will not be important - for other tasks, they will be critical.

Rather than consider the set of "equivalent" sidechain conformations, the standard approach in statistical physics would be to consider the *distribution* over sidechain structures. Any property of interest (e.g. prediction of a downstream task) could then be considered an expectation under this distribution. For instance, if we observe the physical system at some temperature in a fixed volume, the Boltzmann distribution will gives us such a distribution. In the case of this manuscript, the authors are interested in keeping the backbone degrees of freedom fixed - and would thus consider this conditional distribution of sidechain given backbone. The above would be valid for a molecular forcefield parameterizing the energy of a system - but one could presumably make a similar argument for MolProbity: that it induces a probability distribution over valid sidechain conformations, over which you then calculate the expectation. As far as I can see, this would lead to a similar procedure as the one the authors are proposing, but where you use the validity scores of molprobility as an energy rather than considering all sidechains conformations as equivalent. This approach would be easier to defend, because it does not rely on exact equivalences between structural states.

Alternatively, the method could be sold purely as a principled way to do data augmentation, for increased performance in a downstream task.

I would suggest that the authors rewrite "1. introduction" and parts of the "2 Conditional Invariances for Proteins" so that the rotational and translation equivariances of the rigid body case are not presented as equivalent as the conditional invariances that they discuss here. As sketched out above, it would probably be most fruitful if the authors get rid of the concept of "conditional invariance" alltogether, because these invariances are task specific and only approximate - and that they instead present the results in terms of expectations over distributions over sidechain distributions. If the authors stick with "conditional invariance", they should make the limitations of this concept clear - and clarify that the underlying goal is to do meaningful data augmentation in protein systems, and perhaps let that be the driving motivation in the paper.
In short, my main problem with the paper is that the conditional invariances are elevated to something fundamental about the physical system - when in reality it is merely a consequence of the fact that certain details are irrelevant for a particular downstream task.

Finally, for completeness, small fluctuations of the *backbone* structure will have similar properties as the sidechain "invariances" - that they will in many cases be "equivalent" with respect to a downstream prediction task.  It would make sense that the authors mention that they make the simplifying choice of considering only the sidechain conformers in this paper, ignoring this other source of fluctuation.


**Summary Of The Paper:**

The manuscript discusses the need to consider a broader class of invariances in protein modelling than the rotation and translational invariances typically considered. The paper introduces the concept of conditional invariance, defined by transformations that modify the sidechain degrees of freedom of a protein, while leaving the backbone degrees of freedom unaltered. The authors propose using a Markov chain Monte Carlo procedure to generate alternative sidechain conformations during training such that the internal representation in a supervised learning task becomes invariant to the sidechain conformation of the input structure.

**Summary Of The Review:**

While the paper is technically sound, the motivatation for the proposed method is not convincing. Furthermore, the results section does not fully convince me that the complexity of the method (i.e. the MCMC sampling during training) is necessary to achieve the reported gains.

---

> ### Author Response · Authors · 2022-11-19
> **Response to Reviewer W5rQ (Part 1/2)**
>
> We would like to thank the reviewer for their review. We address their concerns below.
>
> **Q1**: The authors introduce the conditional invariances to sidechain conformations as a fundamental property on par with rotational and translation invariances of the global structure of the molecule, but this equivalence seems shaky. it would probably be most fruitful if the authors get rid of the concept of "conditional invariance" altogether, because these invariances are task specific and only approximate - and that they instead present the results in terms of expectations over distributions over sidechain distributions. If the authors stick with "conditional invariance", they should make the limitations of this concept clear - and clarify that the underlying goal is to do meaningful data augmentation in protein systems, and perhaps let that be the driving motivation in the paper. In short, my main problem with the paper is that the conditional invariances are elevated to something fundamental about the physical system - when in reality it is merely a consequence of the fact that certain details are irrelevant for a particular downstream task. Alternatively, the method could be sold purely as a principled way to do data augmentation, for increased performance in a downstream task.
>
> **A1**: Thank you very much for bringing this up.  We largely agree with the reviewer on the points listed here - but look at these as features of our learning procedure, as opposed to a limitation - which we detail next.
>
> 1. Conditional Invariances are task specific? -- While in our work, as we employ a supervised learning paradigm to learn invariances, and therefore this equates to learning task specific invariances (which is more suited here), it is definitely possible to learn conditional invariances which are more general and not limited to a task using a purely self supervised learning paradigm with contrastive learning. We do however agree, that in the case of proteins, conditional invariances may not always be as fundamental as invariances to rotations and translations - we have now updated the manuscript to appropriately convey this [including introduction]. However, in more general settings outside that of proteins, conditional invariances can be fundamental -- for example - different humans have different flexibility (gymnasts for instance, in general, are more flexible than other individuals) and this flexibility has benefits which are not restricted to competitive settings.
>
> 2. Clarify that the underlying goal is to do meaningful data augmentation in protein systems -- We would like to clarify to the reviewer that our model is indeed an augmentation mechanism, akin to Murphy, et al. 2019, Finzi, et al 2020 (As we have appropriately cited in the paper) - but unlike the previous two uses a principled MCMC procedure along with it - which to the best of our knowledge has been unexplored. The Monte Carlo procedure as proposed in Murphy, et al. 2019, Finzi, et al 2020 forms the basis of our ablation study in the appendix - which we outperform. We have updated the manuscript to reflect this as well.
>
> **Q2**: It would make sense that the authors mention that they make the simplifying choice of considering only the sidechain conformers in this paper, ignoring this other source of fluctuation. The result section should include a baseline where the MCMC is done offline, rather than as part of the training procedure. If a pool of samples could be generated prior to training and used as standard data augmentation (e.g. sampling randomly from the pool in each epoch), it would simplify the training procedure considerably.
>
> **A2:** To clarify, as a part of our procedure, we sample a single conformation from the Markov Chain for each protein in every epoch and perform computations using this sampled protein conformation - everything else remaining the same. We do not expect any difference between our learning procedure and that of performing MCMC offline - and preliminary experiments from our end showcase the same (We will continue doing these experiments and add them as updates in comments until the Dec 12 private rebuttal deadline). The other ablation study that we performed (in appendix) is where we obtain new conformations using a Monte Carlo procedure, where in every epoch we start from the original conformation in the dataset rather than from previously made changes (i.e. MCMC is restarted from the original conformation at every epoch - 100% restart - while in the main procedure it uses the changes made to the protein structure in the previous epoch).

---

> > ### Author Response · Authors · 2022-11-19
> > **Response to Reviewer W5rQ (Part 2/2)**
> >
> > **Minor Concerns:**
> >
> > **Concern 1:** Typos: Thank you for pointing these out. We have corrected the typos in the updated manuscript.
> >
> > **Concern 2:** "and the obtained structure is normally considered a unique 3D conformation of the molecule." It is unclear what "unique" means here". Do you mean "representative"?
> >
> > **A:** Thank you, yes we do mean a representative (commonly incorrectly perceived as unique conformation) of the protein.
> >
> > **Concern 3:** Proposition 3.3 and Proposition 3.4 --  The authors should state if these results are any different from the general results known for MCMC simulations of proteins. I would think that it is well known that transitions kernels such as the one described would lead to a Markov chain with a unique steady state distribution. If this is indeed the case, the authors should state that they are simply restating known results. If not, they should state how their result is different.
> >
> > **A**: As we state in the paper, the transition kernels of many other MCMC simulations of proteins (e.g. ) do satisfy Propositions 3.3 and 3.4. And again, as we state in the paper, our goal here was to propose a very simple strategy which would satisfy Proposition 3.3 and 3.4 and yield performance gains without adding excessive computation time (average of 4 minutes additional total training time).
> >
> > **Concern 4:** "Then, a simple way to obtain conformer invariant representations of protein x_j (apart from using trivial functions such as a constant function or function independent of X_v)" What is X_v here?
> >
> > **A** Thank you for bringing this up. We meant X_p here (the atomic coordinates) and have fixed this in the manuscript.
> >
> > Please let us know if you are satisfied with our responses and/ or have any further queries that we can address - and we will add our responses before the end of the Private Reviewer-Author Discussions deadline (12 December - per the Author Guide on ICLR portal). If we have addressed your concerns, please consider revising your score.
> >
> >
> > **References**
> > 1. Murphy, Ryan L., et al. "Janossy pooling: Learning deep permutation-invariant functions for variable-size inputs." ICLR 2019
> > 2. Finzi, Marc, et al. "Generalizing convolutional neural networks for equivariance to lie groups on arbitrary continuous data." International Conference on Machine Learning. PMLR, 2020.

---

> > > ### Comment · Reviewer_W5rQ · 2022-11-22
> > > **Question regarding offline MCMC simulation**
> > >
> > > Thanks to the authors for the clarification regarding the MCMC procedure. My question was perhaps not very clearly phrased: I meant whether it would not be a reasonable baseline to do an offline MCMC simulation and then simply draw one conformation (from the offline samples) in each epoch of training? This would be a much simpler approach - similar to how many implement data augmentation. I was just wondering what the advantages of MCGD were over this simpler approach.

---

> > > > ### Author Response · Authors · 2022-11-25
> > > > **Response to Reviewer**
> > > >
> > > > We would like to thank the reviewer for their clarifying question. In MCGD - the sampling procedure from the time homogenous Markov Chain is independent of the parameters of the NN and hence there is no difference between the mentioned approaches - i.e. sampling from the Markov chain completely first and then training vs a pipelined / interleaved Markov Chain and training procedure - apart from obvious timing benefits.

---

### Official Review · Reviewer_z3cU · 2022-10-24

**Confidence:** 3
**Correctness:** 2
**Technical Novelty And Significance:** 2
**Empirical Novelty And Significance:** Not applicable
**Recommendation:** 3

**Clarity, Quality, Novelty And Reproducibility:**

The efforts of considering possibly non-rigidity of protein conformations besides other group invariance in protein representation learning can be important. However, there are several major concerns on the presented theoretical results. More comprehensive empirical results may be needed to show the significance of the developed procedure. There are also problems in presentation quality as detailed below:

1. The authors should pay careful attention to math notations. For example, on page 3, when introducing the math representation of protein conformers, it is not clear whether the node set is for the number of amino acids or atoms in each amino acid. What did the authors many by  "m-atom" proteins? If the authors directly modeled m atoms, why V={1, ..., n} before Definition 2.2?

2. The notations in the last paragraph and Figure 2 on page 4 are not consistent either. For example, The group of actions is denoted as $G$ in the last paragraph but $\mathcal{G}$ in Figure 2 and its caption. Some notations and acronyms in the paper were not defined clearly, which could be improved for better readability.


**Strength And Weaknesses:**

The authors developed a learning framework that may help protein representation learning considering possible conformation transformations for given input proteins. The procedure of generating and sampling viable conformations is integrated by MCGD to protein representation learning that may help achieve better protein property prediction tasks.

This reviewer has some concerns:

1. The theoretical analyses for the ergodicity and the convergence to the steady-state conformer distributions seems to be problematic. Based on the description of the procedure, the sampled conformers will be validated by "structure validation tools such as Molprobity". Introducing such procedures may violate the desired ergodicity critical in propositions 3.2 and 3.3. Based on the presentation, this reviewer was not sure whether the proposed directed forest construction procedure will always guarantee that all the viable conformers can indeed be included. If not, the procedure does not really have the guarantee to be "conformer invariant" exactly.

2. The theoretical analysis appears to be dependent on Definition 2.2  of "rigid backbone protein conformations" while in the main text and supplement, the authors emphasized that the proposed work is for non-rigid transformations. The authors may clearly state the limitations of the proposed procedure.

3. The presented empirical results are limited with some tasks that the proposed procedure under-performing the baseline without considering "conformer invariance". Also, the proposed procedure appears to share some similarity with augmentation tricks. The authors may want to move the results in the appendix with augmentations into the performance comparison experiments of the main text to check whether the proposed procedure is meaningful or significant. Also, if the authors emphasized that considering more flexible conformers is critical, the performance comparison with other models considering rigid transformations may need to be checked.

**Summary Of The Paper:**

The authors proposed a "conformer invariant representation learning" for protein property prediction in this submission. The basic idea is to sample viable conformation based on the given input protein and conformation. MCMC for sampling conformer distribution is integrated to representation learning via the existing Monte-Carlo Gradient Descent (MCGD) algorithm. The authors have also reported empirical results for four protein property prediction tasks on the ATOM3D data set.

**Summary Of The Review:**

The authors developed a learning framework that may help protein representation learning considering possible conformation transformations for given input proteins. This reviewer has concerns on both reported theoretical and empirical results in the current version as detailed above.

---

> ### Author Response · Authors · 2022-11-19
> **Response to Reviewer z3cU (Part 1/2)**
>
> We would like to thank the reviewer for their review. We address their concerns below.
>
> **Q1:** The theoretical analyses for the ergodicity and the convergence to the steady-state conformer distributions seems to be problematic. Based on the description of the procedure, the sampled conformers will be validated by "structure validation tools such as Molprobity". Introducing such procedures may violate the desired ergodicity critical in propositions 3.2 and 3.3. Based on the presentation, this reviewer was not sure whether the proposed directed forest construction procedure will always guarantee that all the viable conformers can indeed be included. If not, the procedure does not really have the guarantee to be "conformer invariant" exactly.
>
> **A1**: We would like to clarify that indeed all possible conformations can be reached using the directed forest construction and the MCMC procedure. We understand the confusion as there could be cases when a bond is broken during the tree construction and therefore not allowing rotation about the particular bond. However, it is important to note that the two atoms would share a common ancestor in the tree and given that the group of rotations SO(3) is path connected - it would allow for all possible viable conformations through a sequence of alternative actions (More details also in proof of proposition 3.3). With regard to leveraging Molprobity -- we state in the paper [See end of Section 3.1] that the reason Molprobity is used is because it is an easily usable open source tool and is just a means to an end (which still gives considerable performance gains -- with an average increase of just 4 minutes of total training time), and there are other alternative tools/ techniques available.
>
>
> **Q2**: The theoretical analysis appears to be dependent on Definition 2.2 of "rigid backbone protein conformations" while in the main text and supplement, the authors emphasized that the proposed work is for non-rigid transformations. The authors may clearly state the limitations of the proposed procedure.
>
>
> **A2:** We would like to clarify to the reviewer that by rigid body transformations - what we mean are only group transformations such as rotations/ translations/ reflections of the protein as a whole -- which to the best of our knowledge is only what prior works in machine learning for proteins have addressed. Whereas in our work, the only part that we treat as rigid are the backbone atoms of the protein (which is typically the case per studies from scientific empirical literature) and we allow complete flexibility of the side chain as biologically viable.
>
>
> **Q3**: The presented empirical results are limited with some tasks that the proposed procedure under-performing the baseline without considering "conformer invariance". Also, the proposed procedure appears to share some similarity with augmentation tricks. The authors may want to move the results in the appendix with augmentations into the performance comparison experiments of the main text to check whether the proposed procedure is meaningful or significant. Also, if the authors emphasized that considering more flexible conformers is critical, the performance comparison with other models considering rigid transformations may need to be checked.
>
> **A3:**
>
> *Baselines considering rigid transformations:*
>
> We would like to emphasize to the reviewer that the baselines we consider - are indeed rotation (GVP-GNN), rotation and translation invariant (E(n) GNN) and therefore do already consider all rigid body transformations of the protein. The goal of our experimental section is to show that, we can combine our procedure with existing baselines to make them better -- and would like to point out to the reviewer that with proposed addition, we observe up to 26 % improvement over the baseline (17 percentage points) in the tasks without much of a computational overhead - average increase of just 4 minutes of total training time) -- highlighting the importance of factoring in conditional invariances. We also note that our proposed procedure performs worse than the baseline in only 1 out of the 12 scenarios (4 tasks x 3 baselines -- GNNs on LEP task - which is quite an anomaly as observed even by the authors of ATOM 3D dataset - that a model which does not consider atomic positions performs better than models which do).
>
> *Also, the proposed procedure appears to share some similarity with augmentation tricks.*
>
> We would like to clarify to the reviewer that our model is indeed an augmentation mechanism, akin to Murphy, et al. 2019, Finzi, et al 2020 (As we have appropriately cited in the paper) - but unlike the previous two uses a principled MCMC procedure along with it - which to the best of our knowledge has been unexplored . The Monte Carlo procedure as proposed in Murphy, et al. 2019, Finzi, et al 2020 forms the basis of our ablation study in the appendix - which we outperform.

---

> > ### Author Response · Authors · 2022-11-19
> > **Response to Reviewer z3cU (Part 2/2)**
> >
> > **Q4:** Presentation Quality
> >
> > **Concern 1:** Confusion about ‘n’ and ‘m’ about the number of amino acids and number of atoms in the protein
> >
> > **A:** Thank you for pointing this out. ‘m’ is used to denote the number of atoms, and ‘n’ the number of amino acids in the protein. We have updated the manuscript to fix the error.
> >
> > **Concern 2:** The notations in the last paragraph and Figure 2 on page 4 are not consistent either. For example, The group of actions is denoted as in the last paragraph but in Figure 2 and its caption
> >
> > **A:**  Thank you for pointing this out. We have addressed this in the updated manuscript.
> >
> >
> > **Concern 3:** Some notations and acronyms in the paper were not defined clearly, which could be improved for better readability.
> >
> > **A:** We have added a section at the start of the appendix.
> >
> >
> >
> > Please let us know if you are satisfied with our responses and/ or have any further queries that we can address - and we will add our responses before the end of the Private Reviewer-Author Discussions deadline (12 December - per the Author Guide on ICLR portal). If we have addressed your concerns, please consider revising your score.
> >
> >
> >
> > **References**
> > 1. Murphy, Ryan L., et al. "Janossy pooling: Learning deep permutation-invariant functions for variable-size inputs." ICLR 2019
> > 2. Finzi, Marc, et al. "Generalizing convolutional neural networks for equivariance to lie groups on arbitrary continuous data." International Conference on Machine Learning. PMLR, 2020.

---

> > > ### Comment · Reviewer_z3cU · 2022-12-11
> > > **remaining concerns**
> > >
> > > While I appreciate the authors' efforts, I remain concerned about the convergence of involving the MCMC scheme and the limited novelty in the proposed approach, if it is another data augmentation trick. Hence, I will keep my score.

---

> > > > ### Author Response · Authors · 2022-12-11
> > > > **Remaining Concerns**
> > > >
> > > > Dear Reviewer,
> > > >
> > > > Thank you very much once again for your update and taking the time to review. It is rather unfortunate that you have brought up novelty as an issue only today (Dec 11th)  - just hours before the deadline. Additionally, we would like to note that the proof convergence of MCMC has been included in the main paper since submission.
> > > >
> > > > We have tried our best to address all your remaining concerns, but we now feel we are unable to convince you without writing a completely new paper altogether.

---

### Official Review · Reviewer_K2LR · 2022-10-24

**Confidence:** 3
**Correctness:** 3
**Technical Novelty And Significance:** 2
**Empirical Novelty And Significance:** 2
**Recommendation:** 5

**Clarity, Quality, Novelty And Reproducibility:**

The paper was too heavy in introducing concepts and very light in explaining or illustrating these concepts.  Key details were moved to Appendix and the extra space is used to explain details of details, which makes it harder to follow the ideas in the paper. For example in Page 6 MCGD algorithm is not explained at all and the reader was referred to Appendix A.9 and that extra space is used to explain the convergence properties of MCGD. Even the Appendix A.9 does not discuss what MCGD does and only discusses other details about the algorithm. The reader now needs to check the paper from 2018 to see the key ideas of the MCGD algorithm. Only half a page is used for experiments due to page limitation and yet again the reader is referred to Appendix for additional experiments.

There is some novelty in the MCMC based conformation invariant sampling strategy, but it is hard to judge the significance of this novelty given limited empirical evidence.

Minor Corrections:

Please correct cyro-EM as cryo-EM

Remove a in "using directed a forest"

Remove extra to in "converges to to a unique stationary"

How true it is to say that V is the "set" of atoms given that this set does not contain distinct elements, so it can't be a set? V is the set of nodes (not atoms).

"n" was first used to denote the number of amino acids but later on it was used to denote the number of nodes in V, and thus the number of non-distinct atoms.  If m is the number of atoms, then what is n?




**Strength And Weaknesses:**

Strengths:

+ MCMC-based strategy for learning representations that are invariant to conditional transformations

Weaknesses:

- Most technical details are moved to Appendix to save space and the extra space is used to explain details of these details moved to Appendix, which makes it difficult to follow certain sections of the paper.

- Experimental results are included with not much insight. It was not clear whether the improvement over the baseline techniques for four tasks were due to proposed conditional sampling or other aspect of the learning changed by MCGD algorithm.


**Summary Of The Paper:**

The paper introduces an MCMC-based framework that introduces conditional transformations that respect certain 3d structural properties of proteins with the motivation to learn conformation invariant representation.

**Summary Of The Review:**

Although the proposed MCMC based conditional sampling strategy was well thought out, some key concepts were not well explained in the main text. Experiments suggest the proposed sampling strategy improves over several baseline from the literature when these baselines were trained with conformation invariant samples generated with the proposed sampling strategy, but experimental evidence is not very compelling about whether this improvement is due to the networks learning better representations or due to change in some other aspects of the learning process involving MCGD.

---

> ### Author Response · Authors · 2022-11-19
> **Response to Reviewer K2LR**
>
> We would like to thank the reviewer for their review. We address their concerns below.
>
> **Q1**: Most technical details are moved to Appendix to save space and the extra space is used to explain details of these details moved to Appendix, which makes it difficult to follow certain sections of the paper (Specifically about MCGD).
>
> **A1:** We have restructured and added some content to the paper to introduce some details about MCGD in the main paper along with other changes.
>
> **Q2**: Experimental results are included with not much insight and are not compelling. It was not clear whether the improvement over the baseline techniques for four tasks were due to proposed conditional sampling or other aspects of the learning changed by MCGD algorithm.
>
> **A2:**: We would like to clarify that the only change in our experimental evaluation is that we use samples drawn from our MCMC chain for subsequent epochs during training rather than using the same protein 3D structure as input to the model. We would like to point out that just with this change, we observe up to 26 % improvement over the baseline (17 percentage points) in the tasks with minimal computational overhead – average increase of just 4 minutes of total training time – highlighting the importance of factoring in conditional invariances. We also perform an ablation study (in Appendix A.8 for lack of space) where rather than drawing samples from a Markov Chain, we use a Monte Carlo procedure to obtain new conformations from the protein structure (obtained from the dataset). Although this Monte Carlo approach  performs better than the baseline, our proposed approach is clearly superior. We will continue to add other augmentation baselines until the Dec 12 author-review rebuttal deadline (since these experiments with exhaustive hyper-parameter search are time consuming).
>
> **Minor Concerns:**
>
> **Concern 1:** Typos: Thank you for pointing these out. We have corrected the typos in the updated manuscript.
>
> **Concern 2:** How true is it to say that V is the "set" of atoms given that this set does not contain distinct elements, so it can't be a set? V is the set of nodes (not atoms).
>
> **A**: We have updated the manuscript to say a set of nodes rather than atoms. Specifically, we treat every atom as unique (equivalently we assign a unique identifier to every atom thereby making it a set)
>
> **Concern 3:** "n" was first used to denote the number of amino acids but later on it was used to denote the number of nodes in V, and thus the number of non-distinct atoms. If m is the number of atoms, then what is n?
>
> **A**:  Thank you for pointing this out. ‘m’ is used to denote the number of atoms, and ‘n’ the number of amino acids in the protein. We have updated the manuscript to fix the error.
>
>
>
> Please let us know if you are satisfied with our responses and/ or have any further queries that we can address - and we will add our responses before the end of the Private Reviewer-Author Discussions deadline (12 December - per the Author Guide on ICLR portal). If we have addressed your concerns, please consider revising your score.

---

### Author Response · Authors · 2022-11-19
**General Response to all Reviewers**

We would like to thank all the reviewers for their feedback. We have updated the manuscript (all updates in blue) to improve readability, provide clarifications and correct minor typos.

With regard to the general reviewer's concern on reproducibility, we will provide the code (which supports multiple baselines as plug and play) as well as model checkpoints, upon acceptance.

---

### Author Response · Authors · 2022-12-05
**A gentle reminder to All Reviewers**

Dear Reviewers,

This is a just a gentle reminder that the discussion period is ending in a week. We would appreciate it if you could take a look at our responses to individual reviews and updates to the draft. Please let us know if you have any additional questions or concerns, as we would be happy to engage in discussion and address them. Thank you once again for your efforts!

---

### Decision · Program_Chairs · 2023-01-20

**Decision:**

Reject

**Justification For Why Not Higher Score:**

Too many (sometimes severe) points of criticism have been raised and still remained after the rebuttal.

**Justification For Why Not Lower Score:**

N/A

**Metareview: Summary, Strengths And Weaknesses:**

All reviewers agreed that there are many critical aspects concerning the underlying model assumptions (such as the usefulness of the proposed graph decomposition for molecule graphs), the usefulness of the generated structures and the process of validating these structures, the over-all novelty and the experimental validation and comparison. Even the most positive reviewer pointed out that there were "strong assumptions without thorough validation and supported evidence both theoretically and empirically". This over-all impression did not change significantly after the rebuttal and discussion phase, and I also largely agree with most raised points of criticism. Therefore I vote for rejection.